# Calibrated Selective Classification

**Adam Fisch**                                                   *fisch@csail.mit.edu*

**Tommi Jaakkola**                                               *tommi@csail.mit.edu*

**Regina Barzilay**                                              *regina@csail.mit.edu*

*Computer Science and Artificial Intelligence Laboratory (CSAIL)*
*Massachusetts Institute of Technology, Cambridge, MA, 02142, USA.*

**Reviewed on OpenReview:** *https://openreview.net/forum?id=zFhNBs8GaV*

## Abstract

Selective classification allows models to abstain from making predictions (e.g., say "I don't know") when in doubt in order to obtain better effective accuracy. While typical selective models can succeed at producing more accurate predictions on average, they may still allow for wrong predictions that have high confidence, or skip correct predictions that have low confidence. Providing *calibrated* uncertainty estimates alongside predictions—probabilities that correspond to true frequencies—can be as important as having predictions that are simply accurate on average. Uncertainty estimates, however, can sometimes be unreliable. In this paper, we develop a new approach to selective classification in which we propose a method for rejecting examples with "uncertain" uncertainties. By doing so, we aim to make predictions with well-calibrated uncertainty estimates over the distribution of accepted examples, a property we call selective calibration. We present a framework for learning selectively calibrated models, where a separate selector network is trained to improve the selective calibration error of a given base model. In particular, our work focuses on achieving robust calibration, where the model is intentionally designed to be tested on out-of-domain data. We achieve this through a training strategy inspired by distributionally robust optimization, in which we apply simulated input perturbations to the known, in-domain training data. We demonstrate the empirical effectiveness of our approach on multiple image classification and lung cancer risk assessment tasks.[1]

## 1 Introduction

Even the best machine learning models can make errors during deployment. This is especially true when there are shifts in training and testing distributions, which we can expect to happen in nearly any practical scenario (Quionero-Candela et al., 2009; Rabanser et al., 2019; Koh et al., 2021). Selective classification is an approach to mitigating the negative consequences of potentially wrong predictions, by allowing models to abstain on uncertain inputs in order to achieve better accuracy (Chow, 1957; El-Yaniv & Wiener, 2010; Geifman & El-Yaniv, 2017). While such systems may indeed be more accurate on average, they still fail to answer a critical question: how reliable are the uncertainties that they use to base their predictions? Often it can be important to precisely quantify the uncertainty in each prediction via calibrated confidence metrics (that reflect the true probability of an event of interest, such as our model being correct). This is particularly relevant in situations in which a user cannot easily reason about themselves in order to verify predictions; in which there is inherent randomness, and accurate predictions may be impossible; or in which high-stakes decisions are made, that require weighing risks of alternative possible outcomes (Amodei et al., 2016; Jiang et al., 2018; Ovadia et al., 2019).

Consider the task of lung cancer risk assessment, where a model is asked if a patient will develop cancer within the next few years (Aberle et al., 2011). A standard selective classifier that aims to maximize accuracy may

---

[1]Our code is available at `https://github.com/ajfisch/calibrated-selective-classification`.

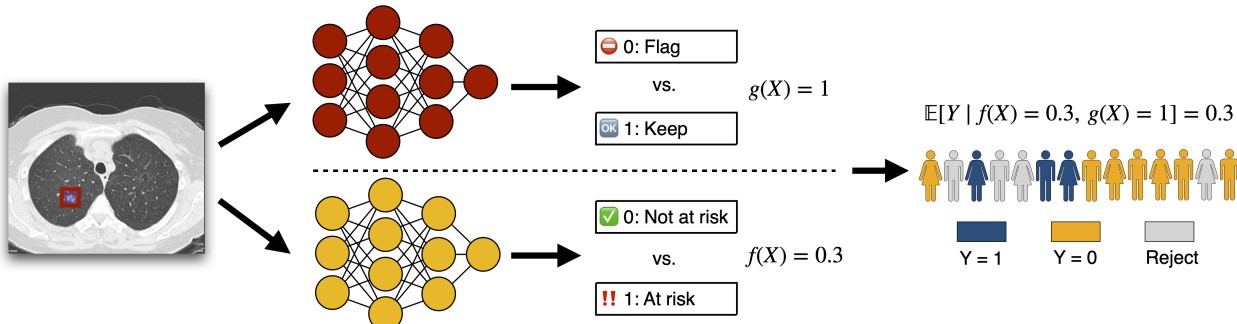

Figure 1: A demonstration of calibrated selective classification for lung cancer risk assessment. The input $X$ is a CT scan; the output $Y$ indicates if the patient develops cancer within 6 years of the scan. $f$ (yellow) provides a confidence estimate for the patient being at risk, while $g$ (red) acts as a gatekeeper to decide when to trust $f$'s outputs. Our goal is for the *accepted* predictions to be calibrated. For example, of all patients with predicted (and non-rejected) risks of 0.3, 30% should be expected to develop cancer within 6 years of the scan.

only make predictions on examples with clear outcomes; predictions that are less useful to the radiologists who these models are meant to help (Dromain et al., 2013; Choy et al., 2018). Furthermore, even if the selective classifier is accurate, it may fail to reliably reflect the risk variations of different patients. For example, suppose that a patient has a 30% chance of developing cancer in reality. The prediction that the patient *will not* develop cancer will most likely be correct. However, knowing the degree of uncertainty in this outcome—namely that the patient's true risk of developing cancer is 30%—might dramatically change the patient's care plan. For these reasons, having a calibrated confidence estimate (in this case, that reflects the true likelihood of developing cancer) can be key to guiding clinical decision making (Jiang et al., 2012; Tsoukalas et al., 2015).

We develop a new approach to selective classification in which we aim to only make predictions with well-calibrated uncertainties. In our lung cancer setting, this means that if the model predicts that cancer will develop with confidence $\alpha$ for a group of patients, then a fraction $\alpha$ of them should indeed develop cancer. Under our framework, a model should abstain if it is uncertain about its confidence, rather than risk misleading a user.[2] In other words, we shift the focus from abstaining on uncertain examples, to abstaining on examples with "uncertain" uncertainties. Figure 1 illustrates our approach.

Concretely, consider a binary classification task (we consider multi-class tasks later) with input space $\mathcal{X}$ and label space $\mathcal{Y} = \{0, 1\}$, where inputs $X \in \mathcal{X}$ and labels $Y \in \mathcal{Y}$ are random variables drawn from a joint distribution $P$. Suppose we have a model $f : \mathcal{X} \rightarrow [0, 1]$, where $f(X)$ represents the model's confidence that $Y = 1$, while $1 - f(X)$ represents the confidence that $Y = 0$. Following the selective classification setting of Geifman & El-Yaniv (2017), we assume that $f$ is *fixed* (e.g., it may be a black-box or expensive to re-train)—and we are now simply seeking a safe gate-keeping mechanism for its predictions. Typical calibration stipulates that $\forall \alpha$ in the range of $f$ we have $\mathbb{E}[Y \mid f(X) = \alpha] = \alpha$, but this may not be satisfied by the provided $f$. Instead, here we let $g \colon \mathcal{X} \rightarrow \{0, 1\}$ be our selection model, where we make a prediction if $g(X) = 1$, and abstain otherwise. Our goal is to then make predictions that are conditionally calibrated, i.e.,

$$\mathbb{E}\big[Y \mid f(X) = \alpha,\ g(X) = 1\big] = \alpha, \quad \forall \alpha \in [0, 1] \text{ in the range of } f \text{ restricted to } \{x : g(x) = 1\}. \quad (1)$$

We refer to this property as *selective* calibration. It is also important to preserve some minimum amount of prediction coverage for $f$ to still be useful (e.g., we may want to make predictions on at least 80% of inputs). As a result, we seek to become as *relatively* calibrated as possible by minimizing the selective calibration error of $(f, g)$, i.e., deviation from Eq. (1), subject to a user-specified coverage constraint on $g$. Inspired by the trainable Maximum Mean Calibration Error (MMCE) objective of Kumar et al. (2018), we propose the *Selective Maximum Mean Calibration Error (S-MMCE)*—along with an efficient method for training $g$ to minimize a (easily estimated) upper bound of it that we compute in practice, in order to improve the selective calibration error. This loss can be effectively trained over a relatively small subset of held-out examples (e.g., $\approx \mathcal{O}(10^3)$).

---

[2]Alternatively, the model may make a *prediction*, but should abstain from providing (or raise a red flag about) its confidence.

Still, the key empirical question that remains, is on what subset of held-out examples can we train $g$? We typically do not know the test distribution, $P = P_{\text{test}}$, that we may encounter, and our goal is for $g$ to generalize beyond simple in-domain examples (for which standard, non-selective, calibration is more readily obtained). To address this, we formulate a robust calibration objective using synthetically constructed dataset shifts. Specifically, given any original training data $\mathcal{D}_{\text{train}} \sim P_{\text{train}}$ that we *do* have, and some family $\mathcal{T}$ of perturbations that we can define (e.g., via common data augmentation techniques), we minimize the S-MMCE over the *uncalibrated* datasets that we create by applying perturbations in $\mathcal{T}$ to the data in $\mathcal{D}_{\text{train}}$. Of course, we do not expect these augmentations to represent all possible test distributions $P_{\text{test}}$. Rather, by constructing a broad, albeit still limited, set of example shifts, our intention is for $g$ to learn to generalize well to new perturbation types, beyond those seen during training. (naturally, $g$'s generalization abilities will depend on how feasible it is to create a meaningful perturbation family $\mathcal{T}$). Empirically, when training and testing on *disjoint* sets of both simulated and naturally occurring perturbations on diverse datasets, we demonstrate consistent empirical reductions in selective calibration error metrics relative to typical confidence-based baselines across multiple tasks and datasets.

**Contributions.** To summarize, the main contributions of this work are as follows:

1. We introduce the concept of *selective calibration* as a selective classification objective;
2. We propose a coverage-constrained loss for training selectors to identify well-calibrated predictions;
3. We provide a robust framework for training selectors that generalize to out-of-distribution data;
4. We demonstrate consistent empirical reductions in selective calibration error metrics (e.g., 18-25% reduction in $\ell_2$ calibration error AUC) relative to standard heuristic baselines across multiple tasks and datasets.

## 2 Related work

**Selective classification.** Selective classification, or classification with a reject option, attempts to abstain on examples that the model is likely to get wrong (El-Yaniv & Wiener, 2010; Geifman & El-Yaniv, 2017). The literature on selective classification is extensive (Hellman, 1970; De Stefano et al., 2000; Herbei & Wegkamp, 2006; Cortes et al., 2016a;b; Ni et al., 2019; Geifman & El-Yaniv, 2019). A straightforward and popular technique is to use some confidence measure to select the most certain examples (Cordella et al., 1995; El-Yaniv & Wiener, 2010; Geifman & El-Yaniv, 2017). If the underlying confidence measure is unreliable, however, this approach can perform poorly (Jones et al., 2021). As in our setting, Kamath et al. (2020) consider selective classification in the context of domain shift, though they focus on accuracy, rather than calibration. Approaches to selective classification that build beyond simple accuracy include cost-sensitive selective classification (Bartlett & Wegkamp, 2008; Charoenphakdee et al., 2021), selective classification optimized for experts in-the-loop (Mozannar & Sontag, 2020), and selective classification with fairness constraints (Shah et al., 2021); all of which represent complementary directions to our work. Lin et al. (2022) also considers the possibility of a non-uniform significance over different classes, and provide a set-based selective classifier with controlled class-specific miscoverage rates. Here, we extend selective classification to focus on improving model calibration over non-rejected instances.

**Model calibration.** Calibration has a rich history in machine learning (Brier, 1950; Murphy & Epstein, 1967; Dawid, 1982; Foster & Vohra, 1998; Gneiting et al., 2007). Recently, it has begun to experience a resurgence in the deep learning literature (Kuleshov & Liang, 2015; Kuleshov et al., 2018; Kumar et al., 2019; van Amersfoort et al., 2020; Gupta et al., 2020), partly motivated by observations that modern neural networks can be significantly miscalibrated out-of-the-box (Guo et al., 2017; Ashukha et al., 2020). To start, a number of efforts have focused on how to best define and *measure* calibration, especially in multi-class settings. A common approach (e.g., as taken by Guo et al. (2017)) measures the top-label calibration error, or the probability that the model's top prediction is correct (a definition we adopt). Other methods propose more precise notions of calibration, including calibration that is conditioned on the predicted class (Gupta & Ramdas, 2022), across classes (Kull et al., 2019), for sub-populations or individuals (Hebert-Johnson et al., 2018; Zhao et al., 2020), or with respect to decision making (Zhao et al., 2021). Additionally, conformal prediction provides tools for calibrating set-based predictions such that they provably satisfy fixed risk limits (e.g., cover the correct label) with some specified probability (Vovk et al., 2005; Bates et al., 2020; Angelopoulos et al., 2022). In the case of providing calibrated *probability* estimates, conformal predictors can also be modified to output (sets of) probability predictions with the same coverage properties (Vovk & Petej, 2014; Vovk et al., 2018). We note that aspects of these methods are complementary to our work, and could be incorporated in a selective setting

following, or similar to, our framework. Finally, approaches to *achieving* better calibration over single point estimates include both parameteric and non-parametric post-processing techniques (Platt, 1999; Zadrozny & Elkan, 2001; Niculescu-Mizil & Caruana, 2005; Guo et al., 2017; Kumar et al., 2019), as well as modified losses introduced during model training (Kumar et al., 2018; Mukhoti et al., 2020; Karandikar et al., 2021). Here, we attempt to achieve better conditional calibration across domains through selective classification.

**Robustness to domain shift.**  It is well-known that model performance can often degrade when the testing and training distributions are different—and many methods have been proposed to help combat this poor behavior (Sugiyama & Müller, 2005; Wen et al., 2014; Reddi et al., 2015; Lipton et al., 2018; Tran et al., 2022). With respect to model calibration, even models that are well-calibrated in-domain can still suffer from considerable miscalibration under domain shift (Ovadia et al., 2019; Minderer et al., 2021). Given some knowledge or assumptions about the possible new target distribution (e.g., such as an estimated likelihood ratios or maximum divergence) several works have considered corrections for calibration under distribution shift (Tibshirani et al., 2019; Cauchois et al., 2020; Gupta et al., 2020; Park et al., 2022). Similarly, given samples from two distributions (e.g., the training and testing distributions), many tests exist for detecting if they are different (Vovk et al., 2003; Gretton et al., 2012; Chwialkowski et al., 2015). A shift in input features, however, is neither necessary nor sufficient as a predictor of accuracy; methods such as Ginart et al. (2022) attempt to monitor when distribution drifts are severe enough to indeed cause a performance drop, in order to trigger a decision to collect more supervised data. Closer to our work, a number of approaches have also attempted robust training (e.g., for temperature scaling) over multiple environments or domains (Wald et al., 2021; Yu et al., 2022), rather than relying on test-time adjustments. Our work adds to the broad array of tools in this area by providing a robust *selective* classification mechanism with few assumptions or requirements.

**Synthetic perturbations.**  Data augmentation is commonly used to improve model performance and generalization (Hendrycks et al., 2020; Rusak et al., 2020; Cubuk et al., 2020; Buslaev et al., 2020). Similarly, automated methods have been developed for identifying challenging sub-populations of a larger training set to target bias (Liu et al., 2021; Bao & Barzilay, 2022). Meanwhile, we create perturbed *datasets* by sampling different augmentation types, in order to uncover relationships between types of perturbations and model calibration. Our line of work is a consumer of the other: as new approaches are developed to expand the space of possible perturbations, the more tools our framework has to use for robust training.

## 3  Background

We briefly review selective classification and calibration. We use upper-case letters ($X$) to denote random variables; lower-case letters ($x$) to denote scalars; and script letters ($\mathcal{X}$) to denote sets, unless otherwise specified.

### 3.1  Selective classification

Returning to the setting in §1, let $f\colon \mathcal{X} \to \mathcal{Y}$ be our prediction model with input space $\mathcal{X}$ and label space $\mathcal{Y}$, and let $g\colon \mathcal{X} \to \{0,1\}$ be a binary selection function over the input space $\mathcal{X}$. Let $P$ be a joint distribution over $\mathcal{X} \times \mathcal{Y}$. For now, we make no distinction between $P_{\text{train}}$ and $P_{\text{test}}$, and simply assume a general $P = P_{\text{train}} = P_{\text{test}}$. A selective classification system $(f,g)(x)$ at an input $x \in \mathcal{X}$ can then be described by

$$(f,g)(x) := \begin{cases} f(x) & \text{if } g(x) = 1, \\ \text{``abstain''} & \text{otherwise.} \end{cases} \tag{2}$$

Selective classifier performance is commonly evaluated in terms of risk versus coverage (El-Yaniv & Wiener, 2010). Coverage is defined as the probability mass of the region of $\mathcal{X}$ that is not rejected, $\mathbb{E}[g(X)]$. In practice, coverage is usually tunable via a threshold $\tau \in \mathbb{R}$, where given some soft scoring function $\tilde{g}\colon \mathcal{X} \to \mathbb{R}$, $g$ is defined as $g(x) := \mathbf{1}\{\tilde{g}(x) \geq \tau\}$. Given some loss function $\mathcal{L}$, the selective risk with respect to $P$ is then defined as

$$\mathcal{R}(f,g;P) := \mathbb{E}[\mathcal{L}(f(X),Y) \mid g(X) = 1] = \frac{\mathbb{E}[\mathcal{L}(f(X),Y)g(X)]}{\mathbb{E}[g(X)]}. \tag{3}$$

$\mathcal{L}$ is typically the 0/1 loss, making $\mathcal{R}(f,g;P)$ the selective error. As evident from Eq. (3), there is a strong dependency between risk and coverage. Rejecting more examples can result in lower selective risk, but also lower coverage. The risk-coverage curve, $\mathcal{R}(f,g;P)$ at different $\mathbb{E}[g(X)]$, and its AUC provide a standard way to evaluate models. We will also use the AUC to evaluate the *calibration*-coverage curve of our proposed method.

### 3.2 Model calibration

We consider two simple marginal measures of binary and multi-class calibration error. Starting with the binary setting, let $f: \mathcal{X} \to [0,1]$ be our model, where $f(X)$ is the confidence that $Y = 1$. The binary calibration error measures the expected difference between the estimate $f(X)$, and the true probability of $Y$ given that estimate.

**Definition 3.1** (Binary calibration error). *The $\ell_q$ binary calibration error of $f$ w.r.t. $P$ is given by*

$$\text{BCE}(f; q, P) := \left( \mathbb{E}\left[ |\mathbb{E}[Y \mid f(X)] - f(X)|^q \right] \right)^{\frac{1}{q}} \tag{4}$$

Typical choices for the parameter $q \geq 1$ are 1, 2, and $\infty$ (we will use both $q = 2$ and $q = \infty$). We now turn to the multi-class case, where $\mathcal{Y} = \{1, \ldots, K\}$ is a set of $K$ classes, and the model $f: \mathcal{X} \to [0,1]^K$ outputs a confidence score for each of the $K$ classes. A common measure of calibration is the difference between the model's highest confidence in any class, $\max_{y \in [K]} f(X)_y$, and the probability that the top predicted class, $\arg\max_{y \in [K]} f(X)_y$, is correct given that estimate (Guo et al., 2017; Kumar et al., 2019):[3]

**Definition 3.2** (Top-label calibration error). *The $\ell_q$ top-label calibration error of $f$ w.r.t. $P$ is given by*

$$\text{TCE}(f; q, P) := \left( \mathbb{E}\left[ |\mathbb{E}[\mathbf{1}\{Y = \arg\max_{y \in [K]} f(X)_y\} \mid \max_{y \in [K]} f(X)_y] - \max_{y \in [K]} f(X)_y|^q \right] \right)^{\frac{1}{q}} \tag{5}$$

In practice, we measure the *empirical* calibration error, where BCE and TCE are approximated (see Appendix B). Having zero calibration error, however, does not imply that $f$ is a useful model. It is still important for $f$ to be a good predictor of $Y$. For example, the constant $f(X) := \mathbb{E}[Y]$ is calibrated, but is not necessarily accurate. As another measure of prediction quality, we also report the Brier score (Brier, 1950):

**Definition 3.3** (Brier score). *The mean-squared error (a.k.a., Brier score) of $f$ w.r.t. $P$ is given by*

$$\text{Brier}(f; P) := \mathbb{E}[(f(X) - Y)^2] \tag{6}$$

In multi-class scenarios, we set $f(X)$ to $\max_{y \in [K]} f(X)_y$ and $Y$ to $\mathbf{1}\{Y = \arg\max_{y \in [K]} f(X)_y\}$.

## 4 Calibrated selective classification

We now propose an approach to defining—and optimizing—a robust *selective* calibration objective. We begin with a definition of selective calibration for selective classification (§4.1). We then present a trainable objective for selective calibration (§4.2), together with a coverage constraint (§4.3). Finally, we construct a framework for training robust selective classifiers that encounter domain shift at test time (§4.4). For notational convenience, we focus our discussion on binary classification, but evaluate both binary and multi-class tasks in §6. As previously stated, we assume that $f$ is a pre-trained and *fixed* black-box, and only train $g$ for now.

### 4.1 Selective calibration

Once more, let $\mathcal{X}$ be our input space and $\mathcal{Y} = \{0, 1\}$ be our binary label space with joint distribution $P$. Again, we assume for now that $P$ is accessible (i.e., we can sample from it) and fixed. When classification model $f: \mathcal{X} \to [0, 1]$ is paired with a selection model $g: \mathcal{X} \to \{0, 1\}$, we define selective calibration as follows:

**Definition 4.1** (Selective calibration). *A binary selective classifier $(f, g)$ is selectively calibrated w.r.t. $P$ if*

$$\mathbb{E}\big[Y \mid f(X) = \alpha, \ g(X) = 1\big] = \alpha, \quad \forall \alpha \in [0, 1] \text{ in the range of } f \text{ restricted to } \{x : g(x) = 1\}. \tag{7}$$

Similarly, we also define the selective calibration error, which measures deviation from selective calibration.

**Definition 4.2** (Selective calibration error). *The $\ell_q$ binary selective calibration error of $(f, g)$ w.r.t $P$ is*

$$\text{S-BCE}(f, g; q, P) := \left( \mathbb{E}\left[ \big|\mathbb{E}[Y \mid f(X), \ g(X) = 1] - f(X)\big|^q \mid g(X) = 1 \right] \right)^{\frac{1}{q}}. \tag{8}$$

---

[3]As discussed in §2, many different definitions for multi-class calibration have been proposed in the literature. For simplicity, we focus only on the straightforward TCE metric, and leave extensions for more precise definitions to future work.

The top-label selective calibration error for multi-class classification is defined analogously, as is the selective Brier score. What might we gain by selection? As a basic observation, we show that the coverage-constrained selective calibration error of $(f, g)$ can, at least in theory, always improve over its non-selective counterpart.

**Claim 4.3** (Existence of a good selector). *For any fixed $f : \mathcal{X} \to [0, 1]$, distribution $P$, and coverage $\xi \in (0, 1]$, there exists $g : \mathcal{X} \to \{0, 1\}$ such that (i) $\mathbb{E}[g(X)] \geq \xi$, and (ii) S-BCE$(f, g; q, P) \leq$ BCE$(f; q, P)$.*

See Appendix A for a proof. Note that in some (e.g., adversarially constructed) cases we may only be able to find trivial $g$ (i.e., the constant $g(X) = 1$ which recovers BCE$(f; q, P)$ exactly), though this is typically not the case. For intuition on how an effective selection rule may be applied, consider the following toy example.

**Example 4.4** (Toy setting). Assume that input $X = (X_1, X_2) \in [0, 1] \times \{0, 1\}$ follows the distribution $P_X := \text{Unif}(0, 1) \times \text{Bern}(p)$ for some $p \geq \xi \in (0, 1)$, and output $Y \in \{0, 1\}$ follows the conditional distribution

$$P_{Y|X} := \begin{cases} \text{Bern}(X_1) & \text{if } X_2 = 1, \\ \text{Bern}(\min(X_1 + \Delta, 1)) & \text{if } X_2 = 0, \end{cases} \tag{9}$$

for some $\Delta > 0$. Further assume that the given confidence predictor is simply $f(X) = X_1$. That is, by design, $f(X) \mid X_2 = 1$ is calibrated, while $f(X) \mid X_2 = 0$ has an expected calibration error of $\approx \Delta$. Marginally, $f(X)$ will have an expected calibration error of $\approx \Delta(1-p) > 0$. It is then easy to see that defining the simple selector $g(X) = X_2$ will result in a *selectively* calibrated model $(f, g)$ with an expected selective calibration error of 0.

Extending this intuition to more realistic settings, our key hypothesis is that there often exist some predictive features of inputs that are likely to be *uncalibrated* (e.g., when $X_2 = 1$ from Example 4.4) that can be identified and exploited via a selector $g$. In the next sections, we investigate objectives for learning empirically effective $g$.

### 4.2 A trainable objective for selective calibration

Our calibration objective is based on the Maximum Mean Calibration Error (MMCE) of Kumar et al. (2018). Let $\mathcal{H}$ denote a reproducing kernel Hilbert space (RKHS) induced by a universal kernel $k(\cdot, \cdot)$ with feature map $\phi : [0, 1] \to \mathcal{H}$. Let $R := f(X) \in [0, 1]$. The MMCE of $f$ w.r.t. $P$ and kernel $k$ is then

$$\text{MMCE}(f; k, P) := \left\| \mathbb{E}\left[ (Y - R)\phi(R) \right] \right\|_{\mathcal{H}}, \tag{10}$$

where $\|\cdot\|_{\mathcal{H}}$ denotes norm in the Hilbert space $\mathcal{H}$. Kumar et al. (2018) showed that the MMCE is 0 iff $f$ is almost surely calibrated, a fundamental property we call faithfulness. Transitioning to selective classification, we now propose the Selective Maximum Mean Calibration Error (S-MMCE). For notational convenience, define the conditional random variable $V := f(X) \mid g(X) = 1 \in [0, 1]$. The S-MMCE of $(f, g)$ w.r.t. $P$ and kernel $k$ is then

$$\text{S-MMCE}(f, g; q, k, P) := \left\| \mathbb{E}\left[ |\mathbb{E}[Y \mid V] - V|^q \phi(V) \right] \right\|_{\mathcal{H}}^{\frac{1}{q}}. \tag{11}$$

S-MMCE involves only a few modifications to MMCE. The iterated expectation in Eq. (11), however, makes the objective difficult to estimate in a differentiable way. To overcome this, instead of directly trying to calculate the S-MMCE, we formulate and optimize a more practical *upper bound* to S-MMCE,

$$\text{S-MMCE}_{\text{u}}(f, g; q, k, P) := \left\| \mathbb{E}\left[ |Y - V|^q \phi(V) \right] \right\|_{\mathcal{H}}^{\frac{1}{q}} = \left\| \frac{\mathbb{E}[|Y - f(X)|^q g(X)\phi(f(X))]}{\mathbb{E}[g(X)]} \right\|_{\mathcal{H}}^{\frac{1}{q}}. \tag{12}$$

As with MMCE and other kernel-based IPMs (such as MMD, see Gretton et al. (2012)), we can take advantage of the kernel trick when computing a finite sample estimate of S-MMCE$_u^2$ over a mini-batch $\mathcal{D} := \{(x_i, y_i)\}_{i=1}^n$, where $(x_i, y_i)$ are drawn i.i.d. from $P$. Denote $f(x) \in [0, 1]$ as $r$. The empirical estimate is

$$\widehat{\text{S-MMCE}}_u^2(f, g; q, k, \mathcal{D}) := \left( \frac{\sum_{i,j \in \mathcal{D}} |y_i - r_i|^q |y_j - r_j|^q g(x_i) g(x_j) k(r_i, r_j)}{\sum_{i,j \in \mathcal{D}} g(x_i) g(x_j)} \right)^{\frac{1}{q}}. \tag{13}$$

Intuitively, we can see that a penalty is incurred when $x_i$ and $x_j$ with similar confidence values $r_i, r_j$ are both selected, but have $r_i, r_j$ that are misaligned with the true labels $y_i, y_j$. In the remainder of the paper we will informally collectively refer to S-MMCE, S-MMCE$_u$, and the empirical estimate as simply the S-MMCE.

### 4.2.1 Theoretical analysis

We briefly highlight a few theoretical properties of S-MMCE, which are mainly analogous to those of MMCE. See Appendix A for proofs. These results also hold in the multi-class case when following a multi-class-to-binary reduction (as in the TCE). First, we show that S-MMCE is a faithful measure of *selective* calibration.

**Theorem 4.5** (Faithfulness). *Let $k$ be a universal kernel, and let $q \geq 1$. The S-MMCE is then 0 if and only if $(f, g)$ is almost surely selectively calibrated, i.e., $\mathbb{P}(\mathbb{E}[Y \mid V] = V) = 1$, where $V := f(X) \mid g(X) = 1$.*

While faithfulness is appealing, we note that S-MMCE $= 0$ is not always achievable for a fixed $f$ (i.e., no selector $g$ may exist). Next, we show a connection between S-MMCE and the expected selective calibration error.

**Proposition 4.6** (Relationship to S-BCE). *S-MMCE $\leq K^{\frac{1}{2q}}$ S-BCE, where $K = \max_{\alpha \in [0,1]} k(\alpha, \alpha)$.*

As can be expected, lower S-BCE leads to a lower upper bound on S-MMCE (both are 0 when $(f, g)$ is calibrated). Lastly, we show that S-MMCE$_u$, which we always minimize in practice, upper bounds S-MMCE.

**Proposition 4.7** (S-MMCE upper bound). *For any $(f, g)$, $P$, and $q \geq 1$ we have S-MMCE $\leq$ S-MMCE$_u$.*

### 4.3 Incorporating coverage constraints

S-MMCE depends heavily on the rejection rate of $g(X)$. As discussed in §1, we desire $g$ to allow predictions on at least some $\xi$-fraction of inputs. This leads us to the following constrained optimization problem:

$$\underset{g}{\text{minimize}} \quad \text{S-MMCE}(f, g; k, P) \quad \text{s.t.} \quad \mathbb{E}\big[g(X)\big] \geq \xi. \tag{14}$$

A typical approach to solving Eq. (14) might be to convert it into an unconstrained problem with, e.g., a quadratic penalty (Bertsekas, 1996), and then to train a soft version of $g$, $\tilde{g} \colon \mathcal{X} \to [0, 1]$, together with some form of simulated annealing to make it asymptotically discrete (as desired). However, we find that we can effectively change the problem slightly (and simplify it) by training the soft selector $\tilde{g}$ on the S-MMCE loss *without* the denominator that depends on $g(X)$. Instead of explicitly training for a specific $\xi$, we simply use a logarithmic regularization term that prevents $\tilde{g}$ from collapsing to 0 (we will later re-calibrate it for $\xi$), i.e.,

$$\mathcal{L}_{\text{reg}}(f, \tilde{g}; q, k, \mathcal{D}) := \lambda_1 \bigg( \sum_{i,j \in \mathcal{D}} |y_i - r_i|^q |y_j - r_j|^q \tilde{g}(x_i)\tilde{g}(x_j)k(r_i, r_j) \bigg)^{\frac{1}{q}} - \lambda_2 \sum_{i \in \mathcal{D}} \log \tilde{g}(x_i) \tag{15}$$

*Regularized S-MMCE$_u$ loss for training soft $\tilde{g}$.*

where $\lambda_1, \lambda_2 \geq 0$ are hyper-parameters. This also has the advantage of avoiding some of the numerical instability present in Eq. (13) for small effective batch sizes (i.e., when either the batch size is small, or the rejection rate is relatively large). The continuous $\tilde{g}$ is then discretized as $g(x) := \mathbf{1}\{\tilde{g}(x) \geq \hat{\tau}\}$, similar to the selective models in §3.1. $\hat{\tau}$ can conveniently be tuned to satisfy multiple $\xi$ (versus being fixed at training time), as follows. Let $\mathcal{S}$ be a set of $\tilde{g}(x)$ scores over an additional split of i.i.d. *unlabeled* data (ideally, test data). We set $\hat{\tau} := \text{Threshold}(\xi, \mathcal{S})$, where $\text{Threshold}(\xi, \mathcal{S})$ is the largest value that at least $\xi$-fraction of $\mathcal{S}$ is greater than:

$$\hat{\tau} := \text{Threshold}(\xi, \mathcal{S}) := \sup \bigg\{ \tau \in \mathbb{R} \colon \frac{1}{|\mathcal{S}|} \sum_{s \in \mathcal{S}} \mathbf{1}\{s \geq \tau\} \geq \xi \bigg\}. \tag{16}$$

While targeting the lowest viable coverage (i.e., $= \xi$ vs $> \xi$) isn't *necessarily* optimal, we empirically find it to be a good strategy. For large enough $|\mathcal{S}|$, we can further show that $g$ will have coverage $\approx \xi$ with high probability.

**Proposition 4.8** (Coverage tuning). *Let $\mathcal{D}_{\text{tune}} := (X_i, \ldots, X_\eta)$ be a set of i.i.d. unlabeled random variables drawn from a distribution $P$. Let random variable $\hat{\tau} := \text{Threshold}(\xi, \{\tilde{g}(X_i) \colon X_i \in \mathcal{D}_{\text{tune}}\})$. Note that $\hat{\tau}$ is a constant given $\mathcal{D}_{\text{tune}}$. Further assume that random variable $\tilde{g}(X)$ is continuous for $X \sim P$. Then $\forall \epsilon > 0$,*

$$\mathbb{P}\big(\mathbb{E}[\mathbf{1}\{\tilde{g}(X) \geq \hat{\tau}\}] \leq \xi - \epsilon \mid \mathcal{D}_{\text{tune}}\big) \leq e^{-2\eta\epsilon^2}, \tag{17}$$

*where the probability is over the draw of unlabeled threshold tuning data, $\mathcal{D}_{\text{tune}}$.*

---

**Algorithm 1** Robust training for calibrated selective classification.

---

**Definitions** $f$ is the confidence model. $\mathcal{D}_{\text{train}}$ is a sample of available training data. $\mathcal{T}$ is a (task-specific) family of perturbations that can be applied to $\mathcal{D}_{\text{train}}$ to simulate potential test-time domain shifts. $\mathcal{D}_{\text{tune}}$ is an additional sample of *unlabeled* data available to us. If possible, $\mathcal{D}_{\text{tune}}$ should be drawn from the test-time distribution. $\xi$ is the target coverage.

1: **function** TRAIN($f$, $\mathcal{D}_{\text{train}}$, $\mathcal{D}_{\text{tune}}$, $\mathcal{T}$, $\xi$)

2:    ▷ Initialize a soft selector model $\tilde{g}\colon \mathcal{X} \to [0,1]$.          (§4.3)

3:    ▷ Define the binarized selector model, $g(x) := \mathbf{1}\{\tilde{g}(x) \geq \tau\}$, for some $\tau \in \mathbb{R}$ that we will choose.

4:    **for** iter $= 1, 2, \ldots$ **do**          (§4.4)

5:       ▷ Randomly sample a batch of $m$ perturbation functions from $\mathcal{T}$, $\mathcal{B}_{\text{func}} := \{t_i \sim \mathcal{T}\}_{i=1}^{m}$.

6:       ▷ Apply each perturbation function $t_i$ to $\mathcal{D}_{\text{train}}$, obtaining $\mathcal{B}_{\text{shifted}} := \{t_i \circ \mathcal{D}_{\text{train}}\colon t_i \in \mathcal{B}_{\text{func}}\}$.

7:       ▷ Calculate the selective calibration error of $(f,g)$ for $\mathcal{Q}_i \in \mathcal{B}_{\text{shifted}}$ using $\tau = \text{Threshold}(\xi, \{\tilde{g}(x)\colon x \in \mathcal{Q}_i\})$.

8:       ▷ (Optional) Identify the $\kappa$-worst datasets, $\mathcal{Q}_{\pi(i)} \in \mathcal{B}_{\text{shifted}}$, $i = 1, \ldots, \kappa$, sorted by selective calibration error.

9:       ▷ Take a gradient step for $\tilde{g}$ using the $\kappa$-worst $\mathcal{Q}_i$ (*or all, if not L8*), $\frac{1}{\kappa}\sum_{i=1}^{\kappa} \mathcal{L}_{\text{reg}}(f, \tilde{g}; q, k, \mathcal{Q}_{\pi(i)})$, see Eq. (15).

10:       ▷ Stop if $\tilde{g}$ has converged, or if a maximum number of iterations has been reached.

11:    ▷ Set final $\hat{\tau}$ on $\mathcal{D}_{\text{tune}}$, where $\hat{\tau} := \text{Threshold}(\xi, \{\tilde{g}(x)\colon x \in \mathcal{D}_{\text{tune}}\})$, see Eq. (16).    (§4.3)

12:    **return** $g(x) := \mathbf{1}\{\tilde{g}(x) \geq \hat{\tau}\}$

---

To summarize, the basic steps for obtaining $g$ are to (1) train a soft $\tilde{g}$ via Eq. (15), and (2) tune a threshold to obtain a discrete $g$ via Eq. (16). Proposition 4.8 shows that this gives us a model with close to our desired coverage with high probability. Still, performance may start to degrade when distributions shift, which we address next.

### 4.4 Learning robust selectors via simulated domain shifts

Until now, we have treated the distribution $P$ as constant, and have not distinguished between a $P_{\text{train}}$ and a $P_{\text{test}}$. A main motivation of our work is to create selective classifiers $(f, g)$ that can effectively generalize to new domains *not* seen during training, i.e., $P_{\text{train}} \neq P_{\text{test}}$, so that our models are more reliable when deployed in the wild. We adopt a simple recipe for learning a robust $g$ for a fixed $f$ through data augmentation. Let $\mathcal{T}$ be a family of perturbations that can be applied to a set of input examples. For example, in image classification, $\mathcal{T}$ can include label-preserving geometric transformations or color adjustments. $\mathcal{T}$ need not solely include functions of *individual* examples. For example, in lung cancer risk assessment, we randomly resample $\mathcal{D}_{\text{train}}$ according to the hospital each scan was taken in, to simulate covariate shift (without directly modifying the input CT scans).

Let $\mathcal{D}_{\text{train}} := \{(x_i, y_i)\}_{i=1}^{n} \sim P_{\text{train}}$ be a split of available training data (ideally distinct from whatever data was used to train $f$). For a given perturbation $t \in \mathcal{T}$, let $\mathcal{Q}_t := t \circ \mathcal{D}_{\text{train}}$ denote the result of applying $t$ to $\mathcal{D}_{\text{train}}$. Depending on the severity of the perturbation that is used (as measured relative to $f$), predictions on $\mathcal{Q}_t$ may suffer from low, moderate, or high calibration error. Motivated by distributionally robust optimization (DRO) methods (Levy et al., 2020), we optimize selective calibration error across perturbations by (1) sampling a batch of perturbations from $\mathcal{T}$, (2) applying them to $\mathcal{D}_{\text{train}}$ and making predictions using $f$, (3) optionally identifying the $\kappa$-worst selectively calibrated perturbed datasets in the batch when taking $\tilde{g}$ at coverage $\xi$,[4] and then (4) optimizing S-MMCE only over these $\kappa$-worst examples (or all examples otherwise).[5] See Algorithm 1. Our method can be viewed as a form of group DRO (Sagawa et al., 2020), combined with ordered SGD (Kawaguchi & Lu, 2020). A unique aspect of our setup, however, is that our groups are defined over general *functions* of the data (i.e., requiring robustness over a broad set of derived distributions), rather than a fixed set of identified sub-populations of the data. Furthermore, the number of functions applied to the data can be combinatorially large—for example, when defining $\mathcal{T}$ to be comprised of compositions of a set of base transformations.

### 4.5 Selector implementation

We implement $\tilde{g}$ as a binary MLP that takes high-level meta features of the example $x$ as input. Specifically, we define $\tilde{g} := \sigma(\text{FF}(\phi_{\text{meta}}(x)))$, where $\sigma(\cdot)$ is a sigmoid output, FF is a 3-layer ReLU activated feed-forward neural

---

[4]When not optimizing for one $\xi$ (e.g., when sharing $\tilde{g}$ for multiple $\xi$), we measure the selective calibration error AUC.

[5]Not all perturbations cause performance degradation, hence the motivation to only optimize over the top-$\kappa$ worst per batch.

network with 64 dimensional hidden states, and $\phi_{\text{meta}}(x)$ extracts the following assortment of fixed features,[6] many of which are derived from typical out-of-distribution and confidence estimation method outputs:

- **Confidence score.** $\max p_\theta(y \mid x)$, where $p_\theta(y \mid x)$ is the model estimate of $p(y \mid x)$ (i.e., equal to $f(X)$).

- **Predicted class index.** One-hot encoding of $\arg \max p_\theta(y \mid x) \in \{0, 1\}^K$, where $K \geq 2$.

- **Prediction entropy.** The entropy, $\sum_{i=1}^K p_\theta(y_i \mid x) \log p_\theta(y_i \mid x)$, of the model prediction, where $K \geq 2$.

- **Full confidence distribution.** Raw model estimate $p_\theta(y \mid x) \in [0, 1]^K$, where $K \geq 2$.

- **Kernel density estimate.** 1D estimate of $p(x)$ derived from a KDE with a Gaussian kernel.

- **Isolation forest score.** 1D outlier score for $x$ derived from an Isolation Forest (Liu et al., 2008).

- **One-class SVM score.** 1D novelty score for $x$ derived from a One-Class SVM (Schölkopf et al., 2001).

- **Outlier score.** 1D novelty score for $x$ derived from a Local Outlier Factor model (Breunig et al., 2000).

- **kNN distance.** 1D outlier score for $x$ derived from the mean distance to its $k$ nearest training set neighbors.

We also use a subset of the derived features individually as baselines in §5.3. For all derived scores, we use the base network's last hidden layer representation of $x$ as features. If the number of classes $K$ is large (e.g., ImageNet where $K = 1000$), we omit the full confidence distribution and the one-hot predicted class index to avoid over-fitting. All models are trained with $q = 2$ for S-MMCE-based losses. For additional details see Appendix B.

## 5 Experimental setup

### 5.1 Tasks

**CIFAR-10-C.** The CIFAR-10 dataset (Krizhevsky, 2012) contains $32 \times 32 \times 3$ color images spanning 10 image categories. The dataset has 50k images for training and 10k for testing. We remove 5k images each from the training set for validation and perturbation datasets for training $\tilde{g}$. Our perturbation family $\mathcal{T}$ is based on the AugMix framework of Hendrycks et al. (2020), where various data augmentation operations (e.g., contrast, rotate, shear, etc) are randomly chained together with different mixture weights and intensities. Our base model $f$ is a WideResNet-40-2 (Zagoruyko & Komodakis, 2016). The last layer features for deriving $\phi_{\text{meta}}(x)$ are in $\mathbb{R}^{128}$. We temperature scale $f$ after training on clean validation data (we use the same underlying set of validation images for training both $f$ and $\tilde{g}$).We evaluate on the CIFAR-10-C dataset (Hendrycks & Dietterich, 2019), where CIFAR-10-C is a manually corrupted version of the original CIFAR-10 test set. CIFAR-10-C includes a total of 15 noise, blur, weather, and digital corruption types that are applied with 5 different intensity levels (to yield 50k images per corrupted test set split). Note that these types of corruptions are not part of the AugMix perturbation family, and therefore not seen during training, in any form.

**ImageNet-C.** The ImageNet dataset (Deng et al., 2009) contains $\approx 1.2$ million color images (scaled to $224 \times 224 \times 3$) spanning 1k image categories. We use a standard off-the-shelf ResNet-50 (He et al., 2016) for $f$ that was trained on the ImageNet training set. We then simply reuse images from the training set to create separate perturbed validation and training datasets for training $\tilde{g}$, and use the same perturbation family as used for CIFAR-10 above (i.e., based on AugMix). We also use the clean validation data for temperature scaleing. The last layer features for deriving $\phi_{\text{meta}}(x)$ are projected down from $\mathbb{R}^{2048}$ to $\mathbb{R}^{128}$ using SVD. Since the number of classes is large, we omit the predicted class index and full confidence distribution from our selector input features. We evaluate on the ImageNet-C dataset (Hendrycks & Dietterich, 2019), which applies many of the same corruptions as done in CIFAR-10-C. Again, none of these types of corruptions are seen during training.

**Lung cancer (NLST-MGH).** As described in §1, in lung cancer risk assessment, lung CT scans are used to predict whether or not the patient will develop a biopsy-confirmed lung cancer within the 6 years following the scan. In this work we utilize, with permission, data and models from Mikhael et al. (2022), all of the which was subject to IRB approval (including usage for this study). The base model for $f$ is a 3D CNN trained on scans from the National Lung Screening Trial (NLST) data (Aberle et al., 2011). We then use a separate split of 6,282 scans from the NLST data to train $\tilde{g}$. The last layer features for deriving $\phi_{\text{meta}}(x)$ are in $\mathbb{R}^{256}$. As the NLST

---

[6]We opt to use fixed, simple, meta features as most of our training sets are small.

data contains scans from multiple hospitals (33 hospitals in total), we are able to construct a perturbation family $\mathcal{T}$ using randomly weighted resamplings according to the hospital each scan is taken from. We evaluate our selective classifier on new set of 1,337 scans obtained from Massachusetts General Hospital (MGH), taken from patients with different demographics and varied types of CT scans. See Appendix B.3 for additional details.

## 5.2 Evaluation

For each task, we randomly reinitialize and retrain $\tilde{g}$ five times, resample the test data with replacement to create five splits, and report the mean and standard deviation across all 25 trials (# models $\times$ # splits). In order to compare performance across coverage levels, we plot each metric as a function of $\xi$, and report the AUC (starting from a coverage of 5% to maintain a minimum sample size). Our primary metric is the **Selective Calibration error AUC** (lower is better). We report both the $\ell_2$ and $\ell_\infty$ error, written as S-BCE$_2$ and S-BCE$_\infty$, respectively (and, accordingly, S-TCE$_2$ and S-TCE$_\infty$). Our secondary metric is the **Selective Brier Score AUC** (lower is better). Thresholds for establishing the desired coverage are obtained directly from the test data, with the labels removed. Hyper-parameters are tuned on development data held out from the training sets (both data and perturbation type), including temperature scaling. Shaded areas in plots show $\pm$ standard deviation across trials, while the solid line is the mean. AUCs are computed using the mean across trials at each coverage level.

## 5.3 Baselines

**Full model.** We use our full model with no selection applied (i.e., $g(x) = 1$ always). All examples are retained, and the selective calibration error of the model is therefore unchanged from its original calibration error.

**Confidence.** We use the model's raw confidence score to select the most confident examples. For binary models, we define this as $\max(f(x), 1 - f(x))$, and $\max_{y \in [K]} f(x)_y$ for multi-class models. Often, but not always, more confident predictions can also be more calibrated, making this a strong baseline.

**Outlier and novelty detection models.** To test if our learned method improves over common heuristics, we directly use the features from §4.5 to select the examples that are deemed to be the least anomalous or outlying. Specifically, we compare to isolation forest scores, one-class SVM scores, and training set average kNN distance by simply replacing $\tilde{g}$ with these outputs.[7] We then apply the same thresholding procedure as in Eq. (16).

# 6 Experimental results

**Selective calibration error.** Figure 2 reports our main calibration results. In all cases, we observe that optimizing for our S-MMCE objective leads to significant reductions in measured calibration error over the selected examples. In particular, S-MMCE-derived scores outperform all baseline scoring mechanisms in both $\ell_2$ and $\ell_\infty$ Selective Calibration Error AUC, as well as in Selective Brier Score AUC. This is most pronounced on the image classification tasks—which we conjecture is due to the effectiveness of AugMix perturbations at simulating the test-time shifts. Note that the lung cancer task has far less data (both in training and testing) than CIFAR-10-C or ImageNet-C, hence its higher variance. Its perturbation class (reweighted resampling based on the source hospital) is also relatively weak, and limited by the diversity of hospitals in the NLST training set. Despite this, S-MMCE still improves over all baselines. Interestingly, while some baseline approaches—such as rejecting based on one-class SVM scores or isolation forest scores—can lead to *worse* calibration error than the full model without abstention, predictors produced using S-MMCE have reliably lower selective calibration error (per our goal) across all coverage rates $\xi$. The fact that our method significantly outperforms distance-based baselines, such as kNN distance or Isolation Forests, is also encouraging in that it suggests that $g$ has learned behaviors beyond identifying simple similarity to the training set.

**Robustness to perturbation type.** Figure 3 shows a breakdown of Selective Calibration Error AUC across all 15 test time perturbations for CIFAR-10-C (note that Figure 2 shows only averages across all perturbations). Naturally, the reduction in calibration error is more pronounced when there is significant calibration error to begin with (e.g., see Gaussian noise). That said, across all perturbation types, we observe substantial decreases in Selective Error AUC, both relative to the full model without abstentions, and relative to a standard confidence-based selection model. Similar results are also seen for ImageNet-C, which we provide in Appendix C.

---

[7]We omit entropy, KDE, and LOF scores for brevity as they perform similarly or worse compared to the other heuristics.

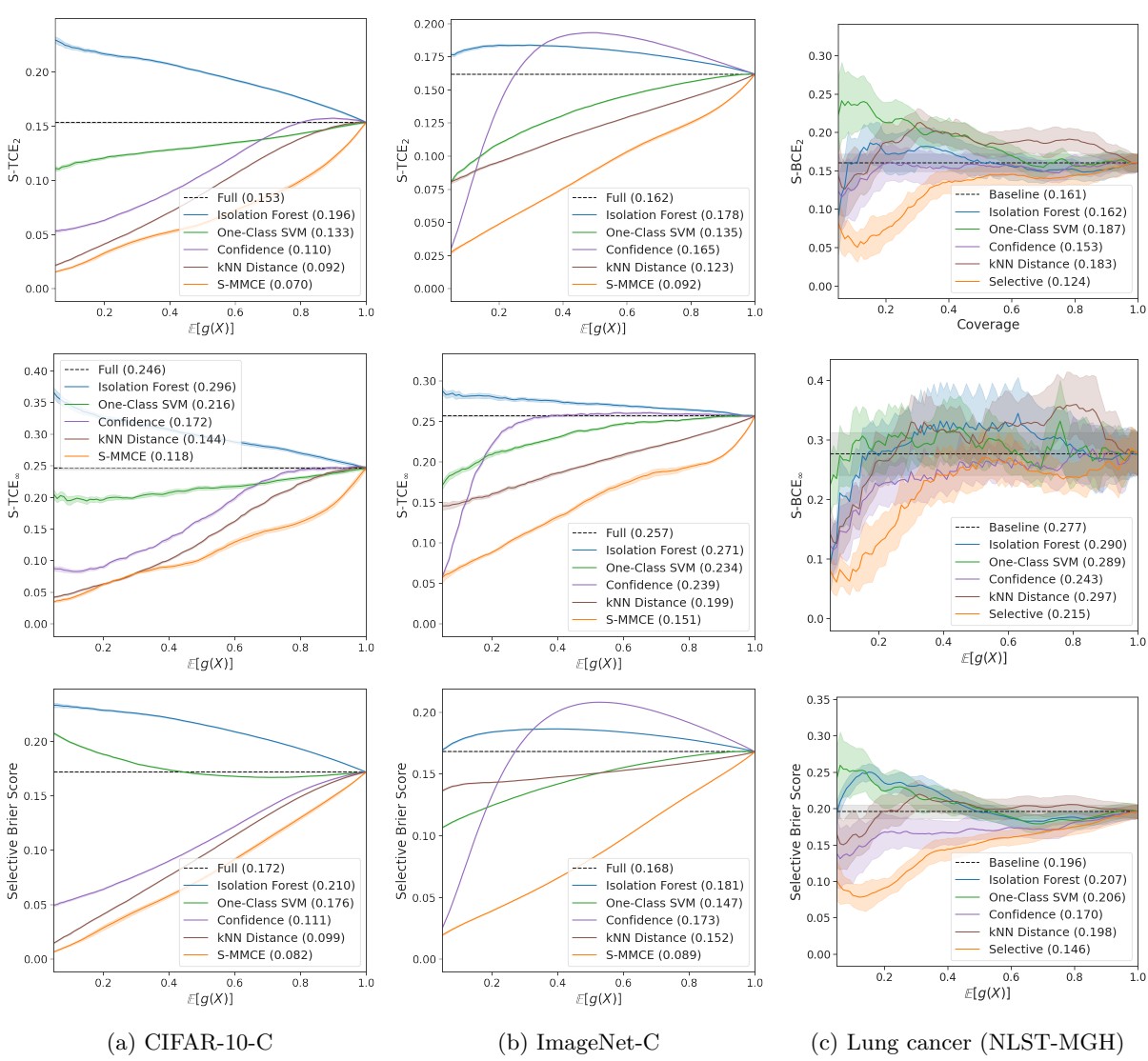

Figure 2: Coverage vs. selective calibration error and Brier scores on CIFAR-10-C, ImageNet-C, and lung cancer data. For CIFAR-10-C and ImageNet-C, we report average results across all perturbation types. For lung cancer, we report results on the diverse MGH test population. Empirically, across all coverage levels and metrics, rejections based on $g$ optimized for S-MMCE perform the best (or on par with the best in some cases).

**Analysis of rejection behavior.** We empirically investigate the types of selections that $g$ trained to optimize S-MMCE make, as compared to typical confidence-based selections. For the lung cancer risk assessment task on MGH data, Figure 4 shows the empirical distribution of confidence scores at various coverage levels $\xi$, conditioned on the example $X$ being selected ($g(X) = 1$). As expected, the confidence-based method selects only the most confident examples, which in this case, is primarily examples where the patient is deemed least likely to be at risk ($f(X) \approx 0$). Figure 4 (b) then demonstrates that an undesirable outcome of this approach is that a disproportionate number of patients that *do* develop cancer ($Y = 1$) are not serviced by the model ($g(X) = 0$). In other words, the model avoids making wrong predictions on these patients, but as a result, doesn't make predictions on the very examples it is needed most. On the other hand, our S-MMCE-based model clearly does not solely reject examples based on raw confidence—the distribution of confidence scores of accepted examples matches the marginal distribution (i.e., without rejection) much more closely. Similarly, as demonstrated in Figure 5, this selector rejects relatively fewer examples where $Y = 1$ as compared to the confidence-based method. Together with the low calibration error results from Figure 2, this suggests that it is more capable of giving calibrated predictions both for non-cancerous and cancerous patients alike.

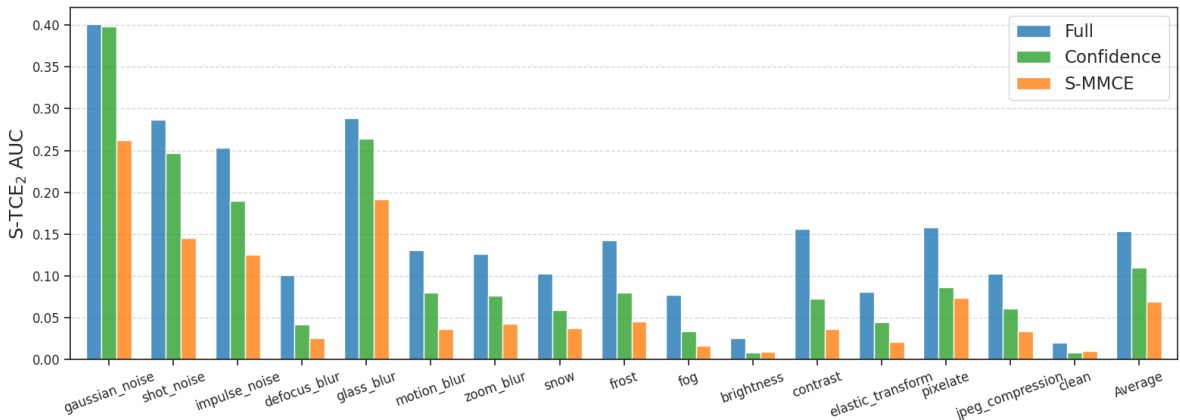

Figure 3: $\ell_2$ selective top-label calibration error AUC reported for each of the 15 test perturbations in CIFAR-10-C (in addition to the average). Optimizing S-MMCE leads to significant error reductions across perturbation types, relative to a model without abstentions ("Full"), as well as the standard selective classifier ("Confidence").

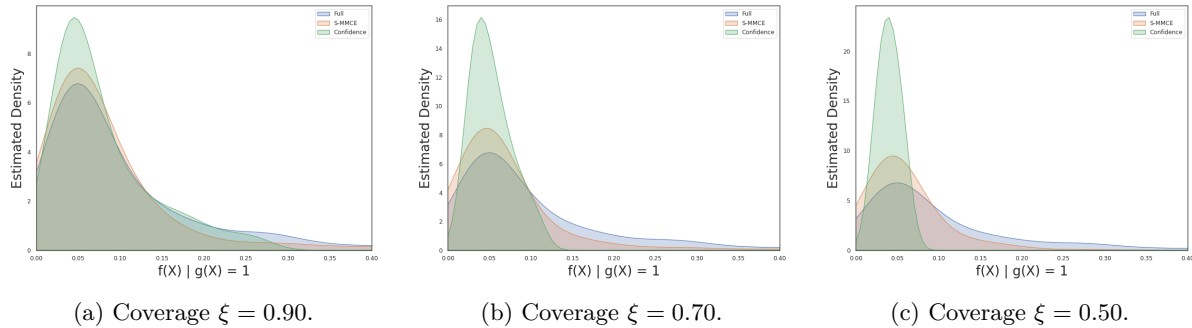

(a) Coverage $\xi = 0.90$.   (b) Coverage $\xi = 0.70$.   (c) Coverage $\xi = 0.50$.

Figure 4: Empirical distribution of $f(X) \mid g(X) = 1$ for different coverage rates on MGH data (for $f(X) \leq 0.4$ for visualization). Empirically, by selecting the most confident predictions, confidence-based predictions mainly take examples that are thought to be *not* be cancerous (i.e., where $f(x) \approx \mathbb{P}(Y = 1 \mid X = x)$ is low). The behavior of the S-MMCE-based classifier, however, is less skewed towards only selecting examples of a particular confidence value, and $f(X) \mid g(X) = 1$ more closely follows the marginal distribution of $f(X)$ without selection.

**Additional results.** We briefly highlight a few additional results that are included in Appendix C. Figure C.3 presents an ablation on CIFAR-10-C that explores the relative importance of different components of our training procedure. Interestingly, even $g$ trained to minimize the S-MMCE on *in-domain* data yield improvements in selective calibration error on out-of-domain data, though the results are significantly improved when including the perturbation family $\mathcal{T}$. Figure C.2 shows the effect of first *training* $f$ on perturbations from $\mathcal{T}$ (though our focus is on black-box $f$). We use a model that is both trained and temperature scaled with AugMix perturbations on CIFAR-10. In line with Hendrycks et al. (2020), this leads to $f(X)$ with lower calibration error on CIFAR-10-C. Still, our framework leads to substantially lower selective calibration error AUC, relative to this new baseline. We also report standard selective *accuracy* scores in Figure C.5. Note that more calibrated classifiers are not necessarily more accurate. Indeed, on ImageNet-C, where S-MMCE yields the most selectively *calibrated*

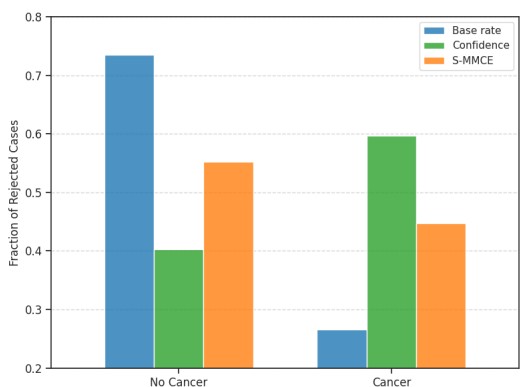

Figure 5: Rejection ratios by label type at $\xi = 0.90$ on MGH data. Blue denotes the ratio of each class in the full data. Proportionally more of the confidence-based rejections are cancerous (presumably as they are "harder" to classify). S-MMCE rejections are relatively less imbalanced.

predictions, it does not yield the best selectively accurate predictions. Still, accuracy is not always at odds with calibration (especially if the higher confidence scores are also calibrated): in CIFAR-10-C and the lung data, S-MMCE also achieves close to the best selective accuracy.

**Limitations and future work.** We note a few considerations that are unaddressed by this work. Marginal calibration does not account for differences between sub-populations or individuals. This is both relevant to fairness and personalized decision making (Zhao et al., 2020; Shah et al., 2021; Jones et al., 2021). While we do not study them here, algorithms for more complete notions of calibration (Hebert-Johnson et al., 2018; Vaicenavicius et al., 2019; Shah et al., 2021), can be incorporated into our framework. We also note that the performance of our method is strongly coupled with the perturbation family $\mathcal{T}$, which, here, is manually defined per task. This can fail when the family $\mathcal{T}$ is not straightforward to construct. For example, the image augmentations such as those in AugMix (Hendrycks et al., 2020) do not generalize across modalities (or even other image tasks, necessarily). Still, as tools for creating synthetic augmentations improve, they can directly benefit our algorithm.

## 7 Conclusion

Reliable quantifications of model uncertainty can be critical to knowing when to trust deployed models, and how to base important decisions on the predictions that they make. Inevitably, deployed models will make mistakes. More fundamentally, not every query posed to even the best models can always be answered with complete certainty. In this paper, we introduced the concept of *selective calibration*, and proposed a framework for identifying when to abstain from making predictions that are potentially *uncalibrated*—i.e., when the model's reported confidence levels may not accurately reflect the true probabilities of the predicted events. We showed that our method can be used to selectively make predictions that are better calibrated as a whole, while still being subject to basic coverage constraints. Finally, our experimental results demonstrated that (1) our calibrated selective classifiers make more complex abstention decisions than simple confidence-based selective classifiers, and (2) are able to generalize well under distribution shift.

## Acknowledgements

We thank Dmitry Smirnov, Anastasios Angelopoulos, Tal Schuster, Hannes Stärk, Bracha Laufer-Goldshtein, members of the Regina Barzilay and Tommi Jaakkola research groups, and the anonymous reviewers for helpful discussions and feedback. We also thank Peter Mikhael for invaluable support on the lung cancer risk assessment experiments. AF is supported in part by the NSF GRFP and MIT MLPDS.

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

## A  Proofs

### A.1  Proof of Claim 4.3

*Proof.* We prove existence in Claim 4.3 by example. Let random variable $R := f(X)$. Define $h(r) = |\mathbb{E}[Y \mid R = r] - r|$ to be the true calibration error for confidence level $r \in [0, 1]$. Then take $g(x)$ as

$$g(x) := \begin{cases} 1 & \text{if } h(f(x)) \le \hat{\lambda}, \\ 0 & \text{otherwise.} \end{cases} \tag{18}$$

where we define $\hat{\lambda}$ as

$$\hat{\lambda} := \underbrace{\inf \left\{ \lambda \in \mathbb{R} \colon \mathbb{P}(h(R) \le \lambda) \ge \xi) \right\}}_{\xi\text{-th quantile of the calibration error of } f}. \tag{19}$$

We show that this choice of $g$ satisfies parts (i) and (ii) of Claim 4.3.

(i) As $g(X)$ is binary, we have $\mathbb{E}[g(X)] = \mathbb{P}(h(R) \le \hat{\lambda})$. The CDF $\mathbb{P}(h(R) \le \lambda)$ per Eq. (19) is right-continuous with $\sup_\lambda \mathbb{P}(h(R) \le \lambda) = 1$, so that $\hat{\lambda}$ is well-defined and satisfies $\mathbb{P}(h(R) \le \hat{\lambda}) \ge \xi$.

(ii) Let random variable $V := R \mid g(X) = 1$. Then let random variables $E, \tilde{E}$ be defined as

$$E := |\mathbb{E}[Y \mid R] - R|, \tag{20}$$

$$\tilde{E} := |\mathbb{E}[Y \mid V] - V|. \tag{21}$$

For our choice of $g$, we have for all $\lambda \in [0, 1]$

$$\mathbb{P}(\tilde{E} \le \lambda) = \mathbb{P}(E \le \lambda \mid E \le \hat{\lambda}) \le \mathbb{P}(E \le \lambda), \tag{22}$$

giving $\mathbb{E}[\phi(\tilde{E})] \le \mathbb{E}[\phi(E)]$ for all increasing functions $\phi$ by stochastic dominance (Shaked & Shanthikumar, 2007). For $x \ge 0$, $\phi(x) := x^q$ is increasing $\forall q \ge 1$, hence $\mathbb{E}[\tilde{E}^q] \le \mathbb{E}[E^q]$, which implies $(\mathbb{E}[\tilde{E}^q])^{\frac{1}{q}} \le (\mathbb{E}[\tilde{E}^q])^{\frac{1}{q}}$.

Comparing definitions, this is equivalent to S-BCE $\le$ BCE. $\qquad\square$

### A.2  Proof of Theorem 4.5

Our proof follows that of Kumar et al. (2018), differing its treatment of (1) random variables conditioned on $g(X) = 1$, and (2) *non-negative* $\ell_q$ calibration error terms. We begin with the following lemma.

**Lemma A.1.** *Let $X, Y \in [0, 1]$ be non-negative random variables with some joint distribution $P$. Define $\mathcal{M}$ as the integral probability metric,*

$$\mathcal{M}(X, Y; \mathcal{C}) := \sup_{h \in \mathcal{C}} \mathbb{E}[Y \cdot h(X)], \tag{23}$$

*where $\mathcal{C}$ is the space of all continuous, bounded functions $h(x)$ defined over $x \in [0, 1]$. Then,*

$$\mathcal{M}(X, Y; \mathcal{C}) = 0 \iff Y = 0 \text{ almost surely.} \tag{24}$$

*Proof.* Since $Y$ is non-negative and bounded, and all $h \in \mathcal{C}$ are bounded, we can write

$$0 \le \mathcal{M}(X, Y; \mathcal{C}) \le B_{\text{hi}} \cdot \mathbb{E}[Y] < \infty \tag{25}$$

for $B_{\text{up}} := \sup_{h \in \mathcal{C}} \sup_{x \in [0,1]} h(x)$, where $0 < B_{\text{up}} < \infty$, as $\mathcal{C}$ includes functions that are not strictly negative.

Similarly,

$$\infty > \mathcal{M}(X, Y; \mathcal{C}) \geq B_{\text{lo}} \cdot \mathbb{E}[Y] \geq 0 \tag{26}$$

for $B_{\text{lo}} := \sup_{h \in \mathcal{C}} \inf_{x \in [0,1]} h(x)$, where $0 < B_{\text{lo}} < \infty$ as $\mathcal{C}$ includes functions that are strictly positive.

Combining bounds from Eqs. (25) and (25), we have

(i) $\mathbb{P}(Y = 0) = 1 \implies \mathbb{E}[Y] = 0 \implies \mathcal{M}(X, Y; \mathcal{C}) = 0$, and

(ii) $\mathbb{P}(Y = 0) < 1 \implies \mathbb{E}[Y] > 0 \implies \mathcal{M}(X, Y; \mathcal{C}) > 0$.

$\square$

We now proceed to prove Theorem 4.5.

*Proof.* For ease of notation, we define random variables $V := f(X) \mid g(X) = 1$ and $E := |\mathbb{E}[Y \mid V] - V|^q$, where $V$ is the selective confidence and $E$ is the selective calibration error at $V$. First, we have that

$$\text{S-MMCE} = 0 \iff \text{S-MMCE}^q = \left\| \mathbb{E}\big[E \cdot \phi(V)\big] \right\|_{\mathcal{H}} = 0,$$

and second that

$$E = 0 \iff \mathbb{E}[Y \mid V] - V = 0. \tag{27}$$

Therefore, it will suffice to show that

$$\text{S-MMCE}^q = 0 \iff E = 0 \; \textit{almost surely.}$$

We start by defining an alternative (intractable) integral probability metric:

$$\mathcal{M}(E, V; \mathcal{C}) := \sup_{h \in \mathcal{C}} \mathbb{E}\left[E \cdot c(V)\right], \tag{28}$$

where $\mathcal{C}$ denotes the space of all continuous, bounded functions $h(v)$ over $v \in [0, 1]$.

Applying Lemma A.1 gives $\mathcal{M}(E, V; \mathcal{C}) = 0 \iff E = 0$ almost surely.

We now replace $\mathcal{C}$ with the set of functions $\mathcal{F} := \{h \in \mathcal{H} : \|h\|_{\mathcal{H}} \leq 1\}$ in the RKHS with universal kernel $k(\cdot, \cdot)$. Since $k$ is universal, $\mathcal{F}$ is dense in the space of bounded continuous functions with respect to the supremum norm. That is, for every $h \in \mathcal{C}$ and $\epsilon > 0$, there $\exists h' \in \mathcal{F}$ such that $\sup_v |h(v) - h'(v)| < \epsilon$. Using the same arguments as Lemma A.1, we can then immediately show that $\mathcal{M}(E, V; \mathcal{F}) = 0$ iff $E = 0$ almost surely.

Finally, using the reproducing property of the RKHS, we derive the equivalence to S-MMCE$^q$:

$$\begin{aligned}
\mathcal{M}(E, V; \mathcal{F}) &= \sup_{h \in \mathcal{F}} \mathbb{E}\left[E \cdot h(V)\right] \\
&= \sup_{h \in \mathcal{F}} \mathbb{E}\left[E \cdot \big\langle h, k(V, \cdot) \big\rangle_{\mathcal{H}}\right] \\
&= \sup_{h \in \mathcal{F}} \big\langle h, \mathbb{E}[E \cdot \phi(V)] \big\rangle_{\mathcal{H}} \\
&= \left\langle \frac{\mathbb{E}[E \cdot \phi(V)]}{\|\mathbb{E}[E \cdot \phi(V)]\|_{\mathcal{H}}}, \mathbb{E}[E \cdot \phi(V)] \right\rangle_{\mathcal{H}} \\
&= \left\| \mathbb{E}\big[E \cdot \phi(V)\big] \right\|_{\mathcal{H}} \\
&= \text{S-MMCE}^q,
\end{aligned} \tag{29}$$

where $\phi$ is the feature map associated with kernel $k$. $\square$

### A.3 Proof of Proposition 4.6

**Lemma A.2.** *Let $X$ be a random variable. For any family of functions $\mathcal{F}$ defined over the range of $X$,*

$$\sup_{h \in \mathcal{F}} \mathbb{E}[h(X)] \leq \mathbb{E}[\sup_{h \in \mathcal{F}} h(X)] \tag{30}$$

*Proof.* For any fixed $h' \in \mathcal{F}$, we have $h'(X) \leq \sup_{h \in \mathcal{F}} h(X)$, which implies $\mathbb{E}[h'(X)] \leq \mathbb{E}[\sup_{h \in \mathcal{F}} h(X)]$. Since this holds for all $h' \in \mathcal{F}$, $\mathbb{E}[\sup_{h \in \mathcal{F}} h(X)]$ is an upper bound of the set $\{\mathbb{E}[h'(X)] : h' \in \mathcal{F}\}$. $\qquad\square$

**Lemma A.3.** *Let $\mathcal{H}$ be an RKHS with kernel $k$. Then the following holds for all functions $h \in \mathcal{H}$:*

$$h(v) \leq \sqrt{k(v,v)} \|h\|_{\mathcal{H}} \tag{31}$$

*Proof.* Using the reproducing property of the RKHS and the Cauchy-Schwartz inequality, we have that

$$\begin{aligned}
h(v) = \langle h, k(\cdot, v) \rangle_{\mathcal{H}} &\leq \|h\|_{\mathcal{H}} \|k(\cdot, v)\|_{\mathcal{H}} \\
&= \sqrt{k(v,v)} \|h\|_{\mathcal{H}}.
\end{aligned} \tag{32}$$

$\square$

We now proceed to prove Proposition 4.6.

*Proof.* We begin by writing S-MMCE in IPM form as in A.2:

$$\begin{aligned}
\text{S-MMCE}^q &= \mathcal{M}(|\mathbb{E}[Y \mid V] - V|^q, V; \mathcal{F}) \\
&= \sup_{h \in \mathcal{F}} \mathbb{E}\left[|\mathbb{E}[Y \mid V] - V|^q \cdot h(V)\right]
\end{aligned} \tag{33}$$

where $\mathcal{F} := \{h \in \mathcal{H} : \|h\|_{\mathcal{H}} \leq 1\}$.

Applying Lemmas A.2 and A.3,

$$\begin{aligned}
\sup_{h \in \mathcal{F}} \mathbb{E}\left[|\mathbb{E}[Y \mid V] - V|^q \cdot h(V)\right] &\leq \mathbb{E}\left[|\mathbb{E}[Y \mid V] - V|^q \cdot \sup_{h \in \mathcal{F}} h(V)\right] \\
&\leq \mathbb{E}\left[|\mathbb{E}[Y \mid V] - V|^q \cdot \sup_v \sqrt{k(v,v)} \|h\|_{\mathcal{H}}\right] \\
&\leq K^{\frac{1}{2}} \mathbb{E}\left[|\mathbb{E}[Y \mid V] - V|^q\right].
\end{aligned} \tag{34}$$

Comparing definitions, we can write $\text{S-MMCE} \leq K^{\frac{1}{2q}} \text{S-BCE}$. $\qquad\square$

### A.4 Proof of Proposition 4.7

*Proof.* We begin by writing S-MMCE in IPM form as in A.2:

$$\begin{aligned}
\text{S-MMCE}^q &= \mathcal{M}(|\mathbb{E}[Y \mid V] - V|^q, V; \mathcal{F}) \\
&= \sup_{h \in \mathcal{F}} \mathbb{E}\left[|\mathbb{E}[Y \mid V] - V|^q \cdot h(V)\right]
\end{aligned} \tag{35}$$

where $\mathcal{F} := \{h \in \mathcal{H} : \|h\|_{\mathcal{H}} \leq 1\}$.

Fix any $h \in \mathcal{F}$ and $v \in [0,1]$. For notational convenience, let $Z := Y \mid V = v$, and define the function

$$\varphi(Z, v) := |Z - v|^q \cdot h(v) \tag{36}$$

Jensen's inequality yields $\mathbb{E}[\varphi(Z, v)] \geq \varphi(\mathbb{E}[Z], v)$, as $\varphi$ is convex in $Z$ for every constant $v$. Applying the law of iterated expectations then gives $\mathbb{E}[\varphi(Z, V)] \geq \mathbb{E}[\varphi(\mathbb{E}[Z], V)]$. Plugging back into Eq. (36), we have

$$\mathbb{E}[|Y - V|^q \cdot h(V)] \geq \mathbb{E}[|\mathbb{E}[Y \mid V] - V|^q \cdot h(V)]. \tag{37}$$

As all $h \in \mathcal{F}$ are part of the RKHS $\mathcal{H}$, their evaluation functionals are bounded, giving

$$\sup_{h \in \mathcal{F}} \mathbb{E}[|Y - V|^q \cdot h(V)] \geq \sup_{h \in \mathcal{F}} \mathbb{E}[|\mathbb{E}[Y \mid V] - V|^q \cdot h(V)]. \tag{38}$$

Rewriting Eq. (38) as a norm in the RKHS $\mathcal{H}$ completes the proof. □

### A.5 Proof of Proposition 4.8

*Proof.* Our proof is similar to that of Bates et al. (2020). For ease of notation, let $S := \tilde{g}(X)$, where by assumption $S$ is continuous and samples $S_i$, $i = 1, \ldots, \eta$ are i.i.d. For some parameter $\tau \in \mathbb{R}$, let

$$\hat{C}_\eta(\tau) := \frac{1}{\eta} \sum_{i=1}^{\eta} \mathbf{1}\{S_i \geq \tau\}, \quad \text{and} \tag{39}$$

$$C(\tau) := \mathbb{E}[\mathbf{1}\{S \geq \tau\}]. \tag{40}$$

Note that $\hat{C}_\eta(\tau)$ is a random variable depending on $S_1, \ldots, S_n$ while $C(\tau)$ is a constant.

For every $\tau \in \mathbb{R}$ and $\forall \epsilon > 0$, Hoeffding's inequality gives a pointwise bound on $\hat{C}_\eta(\tau)$'s deviation from $C(\tau)$:

$$\mathbb{P}(\hat{C}_\eta(\tau) - C(\tau) \geq \epsilon) \leq e^{-2n\epsilon^2}. \tag{41}$$

Recall the definition of $\hat{\tau}$ from Eq. 16:

$$\hat{\tau} := \sup\left\{ \tau \in \mathbb{R} \colon \hat{C}_\eta(\tau) \geq \xi \right\}. \tag{42}$$

As $\hat{C}_\eta$ is a random variable, $\hat{\tau}$ is also a random variable—as is $C(\hat{\tau})$, the true expected coverage at random threshold $\hat{\tau}$. Let $E$ be the event that $C(\hat{\tau}) \leq \xi - \epsilon$. We will now show that this event occurs with probability at most $e^{-2n^2}$. Define the constant $\tau^*$ as $\tau^* := \sup\{\tau \in \mathbb{R} \colon C(\tau) \geq \xi - \epsilon\}$. $C$ is continuous, as $S$ is continuous by assumption, therefore $\tau^*$ is well-defined, and satisfies $C(\tau^*) = \xi - \epsilon$. Suppose $E$. Then $\hat{\tau} \geq \tau^*$, as $C(\tau)$ is monotonically non-increasing. This implies $\hat{C}_\eta(\tau^*) \geq \xi$, as $\hat{C}_\eta(\tau)$ is also monotonically non-increasing.

As the event $C(\hat{\tau}) \leq \xi - \epsilon$ implies the event $\hat{C}_\eta(\tau^*) \geq \xi$,

$$\mathbb{P}(C(\hat{\tau}) \leq \xi - \epsilon) \leq \mathbb{P}(\hat{C}_\eta(\tau^*) \geq \xi). \tag{43}$$

However, by applying Hoeffding's inequality at $\tau = \tau^*$ we also have that

$$\mathbb{P}(\hat{C}_\eta(\tau^*) \geq \xi) = \mathbb{P}(\hat{C}_\eta(\tau^*) \geq C(\tau^*) + \epsilon) \leq e^{-2n\epsilon^2}. \tag{44}$$

Combining bounds in Eqs. (44) and (43) yields $\mathbb{P}(C(\hat{\tau}) \leq \xi - \epsilon) \leq e^{-2\epsilon^2}$.

□

**Remark A.4.** Alternatively, to guarantee coverage $\geq \xi$ with high probability for any $\eta$ (instead of providing a probabilistic bound on its error), one can also use a corrected threshold via the conformal prediction, RCPS, or CRC algorithms (see Vovk et al., 2005; Bates et al., 2020; Angelopoulos et al., 2022, respectively).

## B Technical details

### B.1 Empirical calibration error

We use equal-mass binning (Kumar et al., 2019; Roelofs et al., 2022) to estimate the expected calibration error. Given a dataset of confidence predictions, $\mathcal{D}_{\text{test}} = \{(f(x_1), y_1), \ldots, (f(x_n), y_n)\}$, we sort and assign each

prediction pair $(f(x_i), y_i)$ to one of $m$ equally sized bins $\mathcal{B}$ (i.e., bin $\mathcal{B}_k$ will contain all examples with scores falling in the $\frac{k-1}{m} \to \frac{k}{m}$ empirical quantiles). If $y$ is binary, the expected binary calibration error is estimated by

$$\text{BCE} \approx \left( \sum_{k=1}^{|\mathcal{B}|} \frac{|\mathcal{B}|_k}{|\mathcal{D}_{\text{test}}|} \left| \frac{\sum_{i \in \mathcal{B}_k} y_i}{|\mathcal{B}_k|} - \frac{\sum_{i \in \mathcal{B}_k} f(x_i)}{|\mathcal{B}_k|} \right|^q \right)^{\frac{1}{q}}. \tag{45}$$

When $y$ is multi-class, we replace $y_i$ and $f(x_i)$ with $\mathbf{1}\{y_i = \text{argmax}_{y \in [K]} f(x_i)_y\}$ and $\max_{y \in [K]} f(x_i)_y$, respectively to compute the TCE. We set the number of bins $m$ to $\min(15, \lfloor \frac{n}{25} \rfloor)$. For selective calibration error, we first filter $\mathcal{D}_{\text{test}}$ to only include the examples that have $g(X_i) = 1$, and then apply the same binning procedure.

## B.2 Training details

We use a Laplacian kernel with width 0.2 in S-MMCE, similar to previous work (Kumar et al., 2018), i.e.,

$$k(r_i, r_j) = \exp\left( \frac{-|r_i - r_j|}{0.2} \right) \tag{46}$$

All models are trained in `PyTorch` (Paszke et al., 2019), while high-level input features (§4.5) for $g$ are computed with `sklearn` (Pedregosa et al., 2011). We compute the S-MMCE with samples of size 1024 (of examples with the same perturbation $t \in \mathcal{T}$ applied), with a combined batch size of $m = 32$. We train all models for 5 epochs, with 50k samples (where one "sample" is a batch $\mathcal{D}$ of 1024 perturbed examples) per epoch ($\approx 7.8$k updates total at the combined batch size of 32). The loss hyper-parameters $\lambda_1$ and $\lambda_2$ were set to $|\mathcal{D}|^{-0.5} = 1/32$ and $1e\text{-}2 \times |\mathcal{D}|^{-1} \approx 1e\text{-}5$, respectively. The $\kappa$ in our top-$\kappa$ calibration error loss is set to 4.

## B.3 NLST-MGH lung cancer data

The CT scans in the NLST data are *low-dose* CT scans from patients enrolled in lung cancer screening programs across the 33 participating hospitals. The MGH data is aggregated across a diverse assortment of all patients in their system who underwent *any* type of chest CT between 2008-2018, and matched NLST enrollment criteria (55-74 years old, current/former smokers with 30+ pack years). Note that since these patients are not enrolled in a regular screening program, not all patients have complete 6-year follow-up records. All patients without documented cancer diagnosis in the 2008-2018 period are assumed to be cancer-free.

## C Additional results

Figure C.1 presents results per perturbation type on ImageNet-C. Like CIFAR-10-C (Figure 3), we see consistent and significant selective calibration error reduction across perturbation types. This is in contrast to the confidence-based selective classifier, which does not always reliably lead to improved selective calibration error. Related to perturbations, a question that arises is what happens if perturbations are also included during the training of $f(X)$? Figure C.2 presents results using an $f$ trained with AugMix perturbations over CIFAR-10, on top of which we learn our $g$ as before (also using the same AugMix perturbation family). While the absolute calibration error of $f$ is lower, in line with Hendrycks et al. (2020), using the selective $g$ further reduces the selective calibration error across all coverage levels. Next, Figure C.3 presents the results of an ablation study on the S-MMCE loss formulation, applied to CIFAR-10-C. Specifically, we look at the effect of not including any perturbations, including perturbations on an example level only (i.e., $t(x)$ rather than $t \circ \mathcal{D}_{\text{train}}$), and using all perturbations in a batch instead of only the $\kappa$-worst. Interestingly, even training the S-MMCE loss only on in-domain, unperturbed data can lead to moderate improvements in selective calibration error. Including perturbations, however, leads to significantly better results. Sharing perturbations across batches or training on the $\kappa$-worst then offer additional, albeit small, improvements. We also perform a similar ablation study on CIFAR-10-C over the features provided to our S-MMCE MLP (§4.5) in Figure C.4. Finally, Figure C.5 demonstrates trade-offs between selective calibration and selective accuracy.

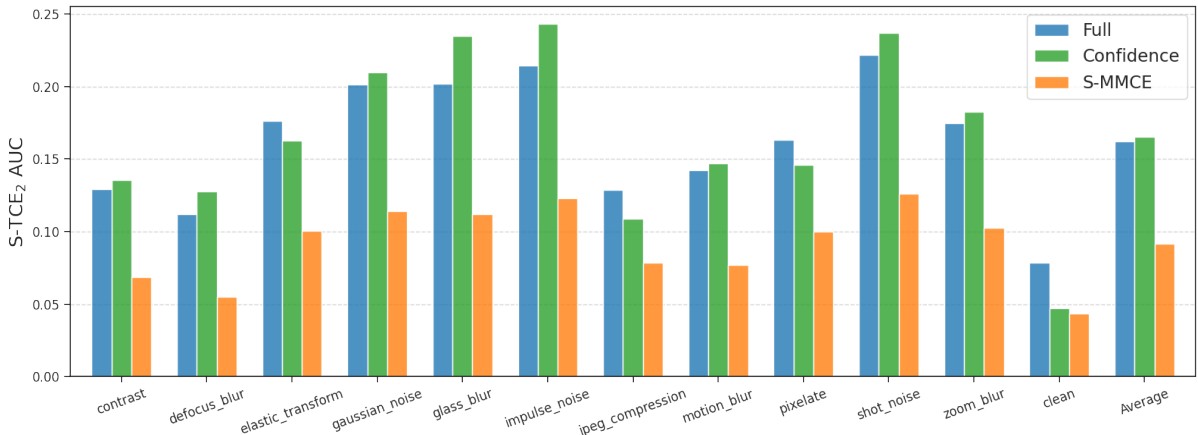

Figure C.1: $\ell_2$ selective top-label calibration error AUC, reported for each of the 11 test-time perturbations in ImageNet-C (and the average). Across perturbations, optimizing for S-MMCE consistently leads to significant error reductions relative to a model without abstentions ("Full"), as well as the standard selective classifier ("Confidence"). In contrast, the confidence-based selector can sometimes lead to worse calibration error.

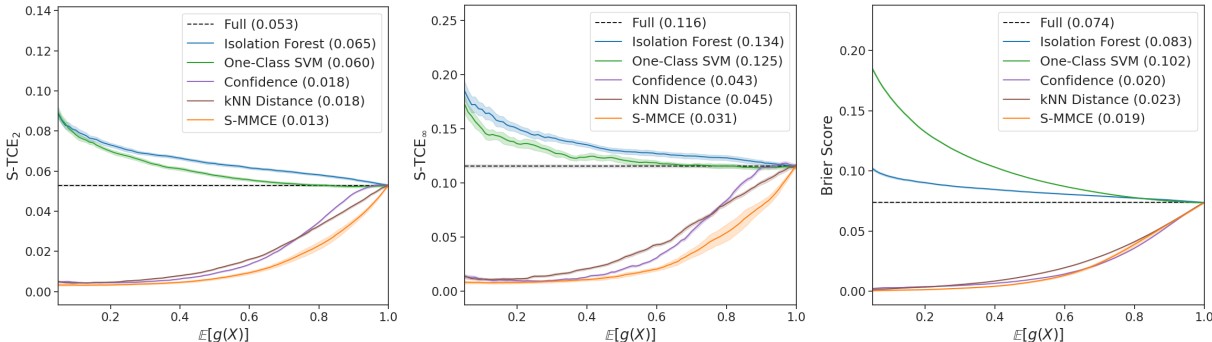

Figure C.2: Results on CIFAR-10-C averaged over perturbation types when including AugMix perturbations ($\mathcal{T}$) during the original training of $f(X)$. As also first reported in Hendrycks et al. (2020), including such perturbations during training indeed reduces calibration error out-of-the-box. Using even the same perturbation class $\mathcal{T}$ during selective training, can yield a selective classifier with substantially lower relative calibration error.

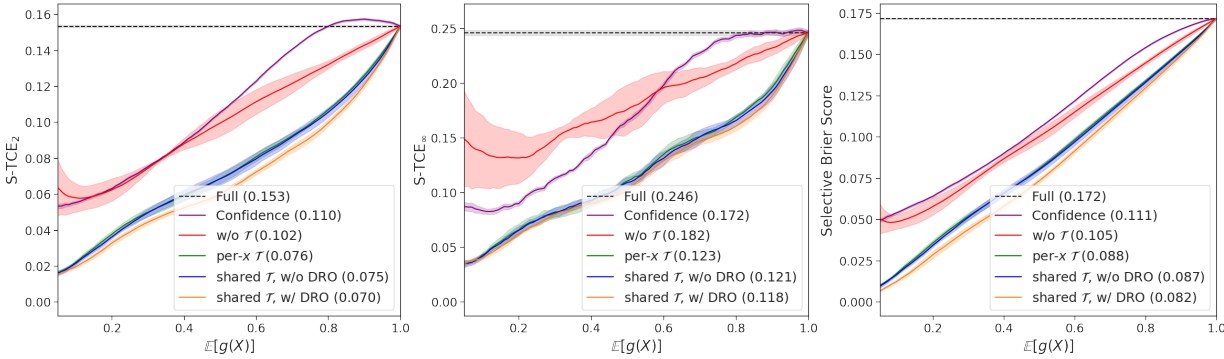

Figure C.3: Ablation on CIFAR-10-C for different configurations of our S-MMCE optimization. Without $\mathcal{T}$: no perturbations. Per-$x$ $\mathcal{T}$: perturbations are mixed within a batch. Shared $\mathcal{T}$, no DRO: update on all batches, not just the $\kappa$-worst. Shared $\mathcal{T}$, with DRO: update on the $\kappa$ worst examples per batch (main algorithm).

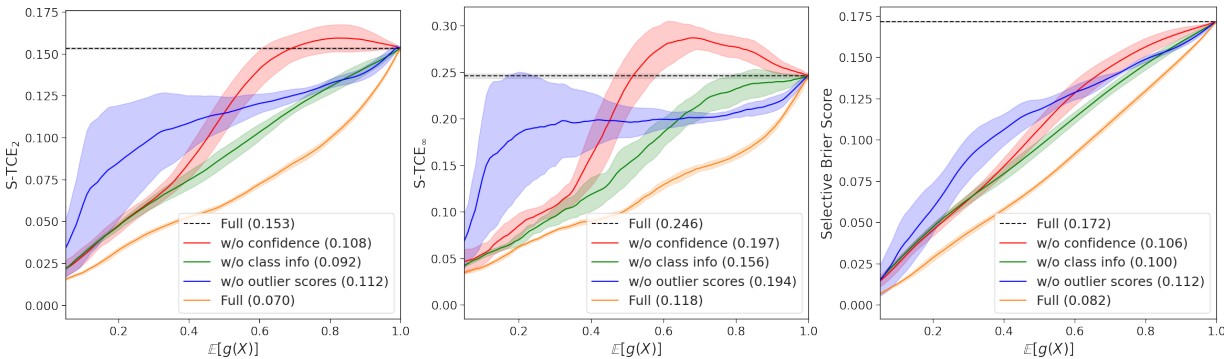

Figure C.4: Ablation on CIFAR-10-C holding out sets of input features to the $g$ MLP. Without confidence: all confidence features are removed (top-label confidence, entropy, the full distribution). Without class info: all features that can be used to identify the predicted class are removed (one-hot class index, the full confidence distribution). Without outlier scores: all derived outlier/novelty scores (One-Class SVM, Isolation Forest, etc) are removed. We see that each of these feature sets has an impact on the aggregate classifier in somewhat complementary ways: in particular, confidence helps performance most at high coverage levels, while outlier scores help performance most at low coverage levels.

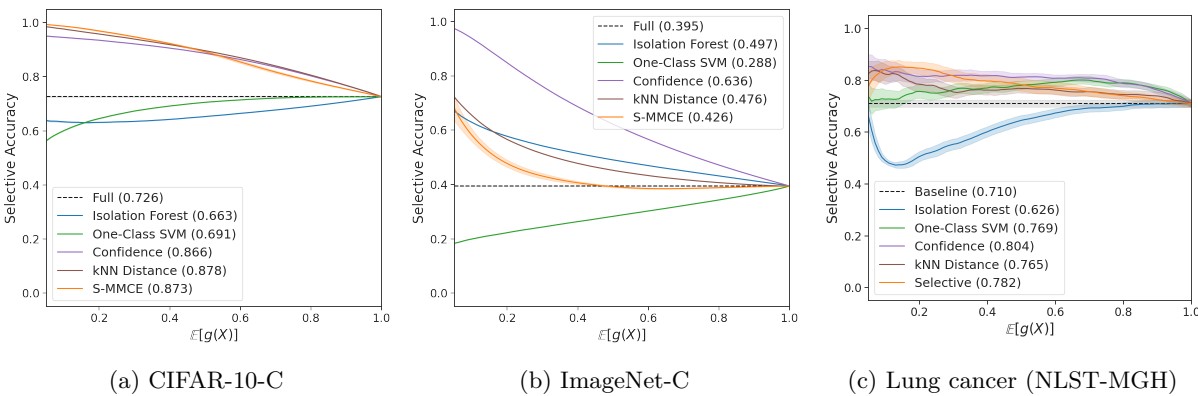

(a) CIFAR-10-C          (b) ImageNet-C          (c) Lung cancer (NLST-MGH)

Figure C.5: Selective accuracy results across CIFAR-10-C, ImageNet-C, and Lung cancer (higher AUC is better). When more calibrated examples also tend to be more accurate (e.g., if higher confidence values are also well-calibrated), then the selective accuracy of S-MMCE can be quite high (see CIFAR-10-C and Lung cancer). Sometimes, however, there is a tradeoff between calibration and accuracy: on ImageNet-C, selection based on S-MMCE scores results in significantly *lower* selective accuracy relative to simple confidence thresholding, but is significantly better calibrated (see Figure 2 in §6).

