# OpenReview forum: "Calibrated Selective Classification"
_TMLR — Accepted by TMLR_

### Review · Reviewer_j41U · 2022-09-27

**Summary Of Contributions:**

* The paper introduces a line of thinking where calibration could be dependent on ‘selection’ of samples, see equation (1).

* Theorem 4.3 provides a proof that such a selectively calibrated classifier exists.

* Measure and loss function are supplied.


**Broader Impact Concerns:**

N.A. Calibration and selective prediction could benefit many other fields of science. Especially enterprises where decisions must be made under uncertainty, such as clinical settings, manufacturing settings, and policy settings.

**Requested Changes:**

 * This paper discusses selective prediction and calibration in this work. However, I am surprised to see no mention of conformal prediction [1]. Could the omission and possible relation be explained? (Same holds for ‘expected calibration error’ and its trainable loss functions [2].)

 * “Under our framework, a model should abstain if it is uncertain about its confidence” This should be motivated further. I would think that ‘Uncertainty’ about a prediction ‘far away from the threshold of g()’ should be considered less than ‘uncertainty’ about a prediction ‘close to the threshold of g()’.


 * Section 4.5: there are important features to use for a g() function. Has research been done to identify the most important features of these? I can imagine some are more important than others. Moreover, the (relative) importance of features could give insight into the working of selective prediction.


 * Most confusing of this paper is the switch between OOD-thinking, like ImageNet-C and Augmix and calibration. Both introduction and conclusion discuss mainly general claims to calibration and selective classification, while experiments relate mostly to OOD datasets and data augmentation


[1] Shafer, G. and Vovk, V., 2008. A Tutorial on Conformal Prediction. Journal of Machine Learning Research.

[2] Guo, C., Pleiss, G., Sun, Y. and Weinberger, K.Q., 2017. On calibration of modern neural networks. In: International conference on machine learning.


**Strengths And Weaknesses:**

Strengths:
  * The paper combines research from selective prediction and calibration, which are two important fields of research.

 * A new tractable metric and corresponding loss function are proposed, whereby selectively calibrated classifiers can be trained/learned.

Weaknesses:

  * The paper seems to be divided between two topics. The introduction, method section and conclusion deal with calibration and selective prediction. Together named selectively calibrated classifiers. And the experimental section deals with OOD tasks. I understand that the two topics are related. But the paper misses either a) experimental results for the new metric (and its loss function), or b) methodical explanation and contributions to the OOD research field are missing.

  * Same question holds for data augmentation, which is discussed in the introduction. Could you please explain in the rebuttal the theoretical connection between selective prediction and calibration on one side, and data augmentation etc on the other side?

  * Training g() requires hold-out data (Section 5.1). Perhaps this weakness should be mentioned earlier on. Moreover, as a question: has research been done to identify the effect of the size of this hold-out data? And what is the effect on more or less held-out data? And finally to play into the experimental section: what is the effect of g() being trained OOD?

  * Section 6: most results focus on OOD performance (ImageNet-C, AugMix). However, I wonder what are the fundamental properties of this new paradigm of ‘selectively calibrated classification’, and properties of the loss function and upper-bound proposed?

* Equation (1): ‘G(x)=1’ seems to be outputting in {0,1} here. However, I assume also the function G() has uncertainty. How is that uncertainty dealt with?

* Again, section 4.4, ‘ A main motivation of our work is to create selective classifiers (f, g) that can effectively generalise to new domains not seen during training‘. I don’t understand how OOD generalisation is a ‘main motivation of the work’.

---

> ### Author Response · Authors · 2022-10-12
> **Response**
>
> We thank the reviewer for their helpful and constructive review!
>
> > On out-of-domain evaluation
>
> Our main motivation is to develop calibrated classifiers that also work in the wild. We hold that in-domain calibration is often somewhat less of an issue, as (a) networks tend to be more calibrated in-domain than out-of-domain [1], and (b) well established methods exist for creating verifiably well-calibrated classifiers on in-domain data, given enough data [2]. We don’t, however, develop our method for applications such as OOD detection—and hence do not claim any contribution to this field. In fact, a well-calibrated selective classifier need not act as an OOD detector at all, if the classifier is still calibrated on the new domain (even if it is not accurate, e.g., if all confidences are low but calibrated). **We have revised parts of the introduction to make this clearer.**
>
> [1] Ovadia et. al. [Can You Trust Your Model’s Uncertainty? Evaluating Predictive Uncertainty Under Dataset Shift](https://arxiv.org/abs/1906.02530).
>
> [2] Kumar et. al. [Verified Uncertainty Calibration](https://arxiv.org/abs/1909.10155).
>
> > On the role of data augmentation
>
> We discuss data augmentation and its use/relation to our work in Section 2. We also informally discuss an interesting connection between our use of data augmentation and a "functional" form of DRO. As in DRO, if our test-time shifts are also in the support of our sampled shifts during training time, then we can expect for our method to do better. In fact, we hypothesize that this is likely the case for why our CIFAR-10-C and ImageNet-C results are relatively so strong (though the test perturbations are still not available at training time, they are still fairly similar, geometric-like perturbations).
>
> Drawing explicit theoretical connections between the type of data augmentation used and the quality of the selective calibration is a very interesting suggestion, but outside of the scope of this work. Here, we measure the effectiveness only empirically.
>
> > On training g()
>
> We are up-front about the data requirements for training g(): we address it in the last paragraph of our introduction. All of the splits we use to train g() are relatively small, in all cases we use splits taken out of original validation sets. The dependence on dataset size also depends on the richness of the perturbation class—a weak perturbation class and a small dataset size could easily lead to a g() that does not generalize well. We have added clarification in the text (see Introduction).
>
> > On fundamental properties
>
> We tried to focus substantial attention on fundamental properties of selective calibration in sections 4.1, 4.2, and 4.3. Our empirical analysis focuses on OOD performance, as we find it most realistic and useful, as per our earlier comment.
>
> > On uncertainty in g()
>
> Indeed, there can be uncertainty in g(), or moreover, uncertainty in the uncertainty in g(), and so forth. We don’t address this here, and only focus on the empirical calibration error. Since we also constrain g() to have a minimal coverage $\xi$, and more rejection generally leads to lower calibration error, we simply threshold g() at the maximum value $\hat{\tau}$ for which $\mathbb{P}(g(X) \geq \hat{\tau}) \geq \xi$ (i.e., the most aggressive it can be). This also goes towards the other question regarding uncertainty close or far away to the prediction threshold (we are as _least_ conservative as allowed).
>
> >  On important features for g()
>
> We have added additional ablations in the Appendix. We see that the different features have an impact on the aggregate classifier in complementary ways: in particular, confidence-related features help performance most at high coverage levels, while outlier/novelty-related scores help performance most at lower coverage levels.
>
> > On conformal prediction and calibration
>
> We have updated our related work section to better highlight additional connections to the conformal prediction literature (note also that previously we had cited Gupta et. al., 2020, Bates et. al., 2020, Angelopoulos et. al., 2020 which address generalized versions of CP). Our proof of coverage preservation is also related to the same concentration inequalities used in RCPS (Bates et. al., 2020), which we cite. As we are aware, typically CP is treated in terms of set- or interval-based predictions with fixed coverage probabilities, whereas we would like low calibration error across point predictions (though we do note Conformal Predictive Distributions and Venn-Abers Predictors have a slightly different flavor, which we also reference). Here we are not so concerned with formal finite-sample guarantees, and only evaluate on the empirical calibration error, which is quite different from the usual goals of CP.
>
> We refer extensively to the calibration literature, including Guo et. al., 2017, in Sections 2/3. Let us know if we missed anything.
>
> Please see our new revision and let us know of any remaining concerns.

---

### Review · Reviewer_ffhS · 2022-10-01

**Summary Of Contributions:**

This work proposes a soft training objective for post-hoc training of selective prediction models, and introduced extensions to ensure the model's out-of-distribution robustness against classes of known and well-defined perturbation functions. On three image classification tasks (CIFAR-C, ImageNet-C and NLST-MGH), the proposed objective achieved superior calibration performance when compared to popular baselines (confidence-based and distance-based outlier detection methods).


**Requested Changes:**

"universal kernel $k(·, ·)$", should it be bound? If so, might be useful to mention it since otherwise

Equation (12): please add a one-sentence explanation about why $S-MMCE_u$ is an upper-bound (maybe just add a pointer to Proposition 4.7).

Equation (13): please comment on the robustness of $S-MMCE_u$ estimator with respect to batch size. I see that there is a possible tradeoff between computational complexity and quality of the S-MMCE estimator. Could the author comment on this possible tradeoff, and provide suggestions on how to decide batch size in practice? A small investigation on one of the dataset would also be very informative.

Proposition 4.7 please discuss how tight this bound is, and if it is not tight, would using $S-MMCE_u$ as the optimization objective impact the calibration quality? Some small experiments on CIFAR-10C / ImageNet-C in comparing the empirical distribution / gap between S-MMCE and S-MMCE_u might be useful.

Small typo: Section 5.1 "ImageNet-C" paragraph: "scaleing -> scaling"

The method's guarantee is currently extended only to the situation where the class of perturbation is known. Authors could consider the method's performance to unknown perturbations (e.g., conduct leave-one-perturbation-out evaluation) to provide a fuller view of the method performance across the datasets. (Just to be clear, even if the method underperforms in this situation, it does compromise the appeal of the method since it is still useful for the situation where all sources of perturbation are known. However, since TMLR insists on the quality and rigor of its publication, I believe presenting an investigation like this to provide a fuller view of the method performance is important).


**Strengths And Weaknesses:**

Strength:
* The paper is well-written, with the objective function clearly introduced and situated in the literature.
* Author derived sufficient theoretical results to backup the design choices
* The method is empirically validated against common baselines on challenging large-scale datasets (ImageNet-C), showing significant improvements.

Weakness: (Please see "Requested Changes")
* The use of S-MMCE_u upper bound is not sufficiently justified, some discussion on either the tightness of the bound, or if the estimation quality compromises the method's objective of minimizing the true S-MMCE (e.g., monitor the S-MMCE_u v.s.  S-MMCE curve during training) would be helpful.
* The S-MMCE_u estimator (Eq. (3)) seems to be numerically unstable depending on the batch size.
* The method's guarantee is currently extended only to the situation where the class of perturbation is known. Authors could consider the method's performance to unknown perturbations (e.g., conduct leave-one-perturbation-out evaluation) to provide a fuller view of the method performance.

---

> ### Author Response · Authors · 2022-10-12
> **Response**
>
> We thank the reviewer for their helpful and constructive review!
>
> > On S-MMCE vs. S-MMCE$_u$
>
> S-MMCE, though useful for deriving S-MMCE$_u$, is not something that we ever explicitly calculated in our experiments. It’s conceivable that one could calculate it, for example, by back-propagating through a sampled, binary g(x) with a straight-through estimator, and then estimating $\mathbb{E}[Y | f(X), g(X) = 1] $with a soft binning scheme, e.g., similar to as done in [1]. Obviously, S-MMCE$_u$ is far more practical to implement—as was the reason for its introduction. Though we can compute S-MMCE for evaluation purposes, we instead focus directly on the more standard calibration error. **We have clarified this in the uploaded revision.**
>
> [1] Karandikar et. al. [Soft Calibration Objectives for Neural Networks](https://arxiv.org/abs/2108.00106).
>
> > On the S-MMCE$_u$ batch size
>
> In our experiments we did not tune the batch size, but rather just took a large batch size that also fit in memory (1024 grouped examples per perturbation, 32 perturbations per batch). We expect only minor variations in performance due to batch size (at least for sizes $> \mathcal{O}(100)$). A large part of this is in fact due to our use of  Eq. (15) vs. Eq. (13). As part of our simplified constrained optimization strategy, Eq. (15) removes the numerically unstable denominator (which depends on batch size/rejection rate), and replaces it with a constant. We have made this point clearer in the uploaded revision.
>
> > On unknown perturbations
>
> **We stress that we do, in fact, evaluate only on unknown perturbations in all of our experiments.** All of the synthetic perturbations in CIFAR-10-C and ImageNet-C are distinct from any of those used during training, see Section 5.1. Furthermore, our MGH testbed for lung cancer risk assessment is from a collection of patients with completely different demographics from the NLST study, with CT scans taken in various formats (e.g., high-dose vs low-dose tomography). Meanwhile, our perturbation class is only defined over the support of the NLST hospitals, see Section 5.1.
>
> > On the choice of kernel
>
> The only constraint we place on the kernel is that it should be a universal kernel—though we are unfamiliar with any standard universal kernels that are unbounded. In practice, we use a Laplace kernel, following prior literature, which is indeed bounded. Using an unbounded kernel will make the connection between ECE and S-MMCE vacuous (Prop. 4.6), although we hold this point to be of only mild theoretical interest.
>
> Please see our new revision and let us know of any remaining concerns.

---

### Review · Reviewer_hv31 · 2022-10-02

**Summary Of Contributions:**

This paper proposes a new learning objective for uncertainty quantification (UQ) called selective calibration, which requires models to abstain predictions based on a selector, so that the remaining predictions are as calibrated as possible (in addition to being accurate). The paper proposes a method for training such a selector, by combining techniques from both the calibration and the distributionally robust optimization literature. The trained selector achieves better selective calibration error over baseline methods.

**Requested Changes:**

Motivation for selective calibration:

The motivation for this task is discussed in the 2nd paragraph in the introduction. While there is an example about cancer development predictions, the example still reads to me like just saying selective classification makes sense, then calibration is also important… so we should do these two together. Could the authors provide more context / examples on who should care about selective calibration? I imagine (and hope!) this paper is going to be read by practitioners, so getting this convincing may be a worthy thing to do for this paper.

Discussions about calibration vs. selective accuracy:

Another point worth some discussions is about the relationship between calibration and selective classification itself. It was observed, e.g. in Malinin et al. (2019, Appendix B & Figure 5) that calibrating the top probability confidence score (which they do by ensemble + distillation) improves the selective classification accuracy defined by the (naive) top probability selector. That’s another connection between calibration and selective accuracy. The learning goal in this paper is different, but I am curious whether the authors think that point is related to this paper / worth to have some discussions.


Reference: Malinin, Andrey, Bruno Mlodozeniec, and Mark Gales. "Ensemble distribution distillation." arXiv preprint arXiv:1905.00076 (2019).


Additional minor questions:

- Brier score reported in Figure 3: Is it actually the *selective* Brier score?
- Choice of the 1d kernel in MMCE (Appendix B.2) is fixed to be a Laplace kernel with $\sigma=0.2$ and not tuned, any intuitions for this choice? (Is it following the literature e.g. the original MMCE paper?)


**Strengths And Weaknesses:**

Strengths:
- The proposed selective calibration objective seems like an interesting addition to the UQ literature. Compared to usual calibration which modifies the confidence scores, the key difference within selective calibration is that it does not directly change the confidence scores, but rather calibrates through a binary decision given by the selector. This may be useful in real-world applications where such selectors are permitted whereas a (full) calibrator is not.

- Experiments seem solid (e.g. the performance improvement, range of baselines and datasets), and justify the advantage of the proposed method against baselines.

- The presentation is very clear and contains lots of useful details.

- A couple design choices are justified by theoretical results, e.g. good property of the population S-MMCE objective when a universal kernel is used. This feels secondary though compared with the above strengths.

Weaknesses:
- My main concern is about *accuracy*: The paper does not report selective accuracy when the proposed calibrated selector is deployed. In Figure 3, only selective calibration objectives like S-TCE or mixed objectives like Brier scores are evaluated.
Selective accuracy itself (e.g. its AUC) is at least as important in my opinion for this task, due to the nature of this problem: The mechanism for selective classification is that there can only be one such selector. It is important to understand whether the calibrated selector is itself a less accurate selector (e.g. compared with SOTA selectors for selective accuracy). Or, if there is a trade-off between selective accuracy vs. selective TCE, where does the proposed method lie on the trade-off curve, and where do the baselines lie.


Given the above strengths and weaknesses, overall, I feel like this paper is a solid UQ/deep learning paper, which proposes a new learning objective and designs a method that achieves it better than a range of baselines. However, there is a concern whether the selective calibration objective is at a cost of other important objectives like selective accuracy.

---

> ### Author Response · Authors · 2022-10-12
> **Response**
>
> We thank the reviewer for their helpful and constructive review!
>
> > On selective accuracy
>
> We appreciate the reviewer’s concern about selective accuracy. However, we emphasize that better calibrated selective classifiers do not necessarily need to be selectively accurate to still be useful (due to the more reliable confidence values that they can provide instead).
>
> Consider our lung cancer risk assessment task. In general, patients that are at risk for developing cancer are relatively harder to accurately predict: after all, developing cancer is a rare event. The most “confident” examples also tend to be ones where the model predicts a very low P(cancer | patient). Since cancer is rare, these predictions might be quite accurate on average. But they either skip the harder cases for when there is in fact something clinical to look for (that, having slightly more inherent uncertainty, can be harder to accurately predict), or include wrong predictions with very high confidence (also undesirable). This can be seen from Figures 4 and 5—”confidence-only” based predictors tend to disproportionately reject cases for which the outcome is $Y=1$ (has cancer). On the other hand, having calibrated confidence scores for $Y=1$ vs $Y=0$ can potentially allow one to take more informed actions, e.g., based on expected risk (even if predicting exactly if $Y=1$ is harder).
>
> We are also slightly unsure of the reviewer's comment:
>
> > The mechanism for selective classification is that there can only be one such selector.
>
> As stated, the focus of this paper is on selective calibration. But, if selective accuracy were also desired, then we don’t see constraints on defining a pair of independent g(x)’s. One to return 1 if the top prediction should be used (for selective accuracy), and the other to return 1 if its confidence should be trusted. Note, by the way, that a benefit of having calibrated classifiers is that the resulting confidence scores can also readily be used as thresholds for selective classification based on accuracy. If we have misunderstood, we will appreciate it if the reviewer can clarify their original statement.
>
> **That said, we have included selective accuracy as another metric in our Appendix (Fig. C.5) and highlighted it briefly at the end of the main results section.** Indeed, we can sometimes see a trade-off (but not always). For example, when the more calibrated scores chosen by S-MMCE also tend to be more accurate (e.g., when higher confidence scores are also not mis-calibrated), S-MMCE results in good selective accuracy (comparable to the best baselines). This is true for CIFAR-10-C and Lung. However, on ImageNet-C—where we see the biggest improvements in calibration error relative to baselines—the selective accuracy is worse than the best baselines.
>
> > On motivation
>
> The use-case of selective calibration is very similar to the use cases of calibrated uncertainty estimates. Calibrated uncertainty estimates may be used for more complex decision making based, for example, on expected risk, as compared to simply taking binary “predict or not” actions. Note that declining to give an uncertainty estimate (as in selective calibration), does not necessarily imply declining to give a raw prediction (see footnote 1 in the paper, and the above discussion).
>
> One such decision making procedure following selective calibration may in fact be standard selective classification—such as making a prediction only if the (non rejected) confidence is $> \alpha$. If simultaneously calibrated across confidence levels, this also implies that the selective accuracy will be $> \alpha$.
>
> Our example in the second paragraph of the Introduction emphasizes that calibration is a different (and potentially opposing) objective than the one typically considered by selective classification (high accuracy). It is not the case that one can simply combine them (and we point out some of the deficiencies of standard selective classification in this paragraph).
>
> > Minor questions
>
> - Yes, it is the selective Brier score (we have updated the figure axes and text).
> - The choice of kernel was borrowed from prior literature (we have updated the description in the implementation section).
>
> Please see our new revision and let us know of any remaining concerns.

---

> > ### Comment · Reviewer_hv31 · 2022-10-24
> > **Response to authors**
> >
> > I thank the authors for the thoughtful response and the revision of the paper. In particular I am glad to see the additional result on selective accuracy, which do show that sometimes (but not always) selective calibration is at odds with selective accuracy. The added sentence in the motivation paragraph was also quite helpful.
> >
> > Re "one such selector": I agree with the rebuttal, you can indeed have two independent selection mechanisms, one for trusted confidence and one for trusted label prediction. I was imagining a different use case where somehow you have to either make this prediction or not make this prediction at all. Also, I think I was indeed

---

### Author Response · Authors · 2022-10-18
**Thank you to reviewers and a request for final comments**

Dear reviewers,

Thank you for your constructive reviews. We have provided our individual responses, and also uploaded an improved draft that addresses the points raised. Let us know if there is anything else that would benefit from further discussion.

Authors

---

### Author Response · Authors · 2022-12-16
**Thank you**

Dear Reviewers and Action Editor,

Thank you very much for your time and efforts spent reviewing and improving our paper. We have uploaded a camera ready version, including links to a public code release. We look forward to future submissions to TMLR.

Authors

---

### Decision · Action_Editors · 2022-11-03

**Recommendation:** Accept as is

**Comment:**

All reviewers are in favour of accepting this work, a viewpoint that I am happy to concur with myself: I believe it is a useful contribution to the uncertainty quantification literature.  As the main concerns of the reviewers have already been addressed in the revision, I am happy to recommend the paper be accepted as is.

**Audience:**

I believe the paper will be of interest to a relatively large number of people in TMLR's audience.

**Claims And Evidence:**

The claims are supported by clear and appropriate evidence.